# Reconstructing meaning from bits of information

Sasa L. Kivisaari [1,2], Marijn van Vliet[1,2], Annika Hultén[1,2], Tiina Lindh-Knuutila[1], Ali Faisal[1] & Riitta Salmelin[1,2]

Modern theories of semantics posit that the meaning of words can be decomposed into a unique combination of semantic features (e.g., "dog" would include "barks"). Here, we demonstrate using functional MRI (fMRI) that the brain combines bits of information into meaningful object representations. Participants receive clues of individual objects in form of three isolated semantic features, given as verbal descriptions. We use machine-learning-based neural decoding to learn a mapping between individual semantic features and BOLD activation patterns. The recorded brain patterns are best decoded using a combination of not only the three semantic features that were in fact presented as clues, but a far richer set of semantic features typically linked to the target object. We conclude that our experimental protocol allowed us to demonstrate that fragmented information is combined into a complete semantic representation of an object and to identify brain regions associated with object meaning.

[1] Department of Neuroscience and Biomedical Engineering, Aalto University, P.O. Box 12200, FI-00076 Aalto, Finland. [2] Aalto NeuroImaging, Aalto University, P.O. Box 12200, Aalto FI-00076, Finland. Correspondence and requests for materials should be addressed to S.L.K. (email: sasa.kivisaari@aalto.fi)

The brain binds available information about objects with prior knowledge, thus allowing us to make sense of the world around us. The ability to use available information about an object (e.g. the observation of something that has legs, is gray and has a trunk) to activate relevant existing knowledge in the semantic system (e.g. is endangered, has white tusks) can be characterized as a process of pattern completion where few elements serve to activate a number of relevant elements in the same representation[1–3]. While one can easily demonstrate the existence of such a process behaviorally, as in the example above, neuroimaging evidence of pattern completion of semantic information is critically lacking. As such, we also do not understand the neuroanatomical bases of this process. Thus, in this study, we ask whether we can demonstrate that semantic pattern completion occurs in the human brain and identify the brain regions which support the representations of object meaning.

Many neurocognitive accounts on the semantic system propose that the meaning of objects can be formalized using smaller components called features[4–7]. The features which make an object are putatively coded in a distributed fashion, primarily in the same regions that are involved in processing and perceiving them[6,8–10]. According to this view, a neural representation of the underlying object would be defined as a specific and relatively stable pattern of activation across the relevant feature nodes[5,7,11]. Computational models further postulate that the activation of a sufficient number of semantic features may lead to activation of the whole semantic representation via a pattern-completion-like process[8,9,11].

Here, we define pattern completion as a partial clue leading into the retrieval of a previously learned memory trace[2,12–14]. Previous research in this field has primarily centered around the recall of episodic memories and the role of the hippocampus in this process[2,12–14]. Several studies have also provided neuroimaging (functional MRI (fMRI)) evidence on the role of the hippocampus in binding partial cues with the context in which they were learned[15–19]. Pattern completion has been demonstrated for example in the visual domain[20], and it has been suggested that pattern completion takes place also for other types of information, including semantic memories[1]. However, there is little we know about the neural underpinnings of pattern completion of semantic memories.

We assess pattern completion of semantic information in the human brain, by making use of a multi-dimensional semantic space. Each dimension in the semantic space corresponds to a single semantic feature. The meaning of an object is defined as a position in this space (a semantic coordinate), which in turn is determined by the weighted combination of the dimensions. The distance (e.g. cosine) between two concepts quantifies their semantic similarity. Semantic spaces can be obtained, e.g. by using statistical co-occurrence information collected from large text corpora[21–24] or using behavioral methods to estimate similarity of descriptive content between items[25–27] (see also ref. [28]). Such semantic spaces have been used as priors in machine-learning-based neural decoding models that have successfully associated various semantic feature sets (i.e. sets of dimensions that span the semantic space) with neural signatures and, by combining them together, predicted neural activation patterns for novel objects[28–34]. This demonstrates that the feature-based model of the semantic system is useful in describing the neural representation of meaningful stimuli.

The visual object processing system may provide insights into the neuroanatomical underpinnings of the semantic pattern completion process. Visual pattern completion has been suggested to take place in the ventral stream via recurrent connections[20]; see also ref. [35]. Particularly, the perirhinal cortex (PRC), which is located in the anterior apex of this hierarchical system, has been deemed relevant in fine-grained visual analysis of

objects[36,37] and binding information across sensory modalities[38,39], including information about object meaning[40]. This region has been suggested to be sensitive to object-specific semantic information[41–43]. Therefore, we hypothesize that the ventral stream system, and the PRC in particular, may be involved in the pattern completion process where fragmental semantic information is completed to form a coherent object.

We probe target objects with a small set of verbal semantic features and thereby putatively facilitate activation in a rich network of other semantic features that are related to the target object. Specifically, we mimic a guessing game where the participant is presented with a sequence of three clues (henceforth, a "clue triplet"; e.g., "has legs", "has a thick skin", "has a trunk") and asked to guess the object that the clues describe (i.e., "an elephant"). We take advantage of functional magnetic resonance imaging (fMRI) and evaluate whether the blood-oxygen-level dependent (BOLD) response is best predicted by the semantic coordinates of the explicitly presented clues or a larger set of features, extending to those that were never presented to the participant (e.g., "is endangered", "is heavy", "does trumpet"). We hypothesize that the brain automatically ties together the presented clues with other features linked to the target object. If so, the best decoding performance would be achieved by using an even larger set of features that are associated with the target object, as compared to the exact selection of features that were presented to the participants.

The semantic space in this study is built from a large Internet-derived Finnish text corpus[22] using the word2vec algorithm[23,44]. In order to predict the brain activity to a given object/feature, we use a linear-regression decoding approach which, for each target object or semantic feature, maps its coordinates in a multi-dimensional semantic space to a corresponding BOLD activation pattern[33,45]. A leave-two-out scheme is used to assess the performance of this mapping. We further employ representational similarity analysis[46] (RSA) to visualize the brain regions which are involved in representing the implied target objects.

We successfully decode object representations from BOLD activity patterns without explicitly showing the target objects. The resulting representations are best decoded by using a rich set of object features, including features that were not presented as clues. These findings demonstrate that the brain uses clues from the environment to build coherent representations of object meaning.

## Results

**BOLD activation patterns reliably predict target objects**. The stimuli in this study consisted of 60 target objects that fell into four semantic categories (animals, fruits/vegetables, tools, vehicles, see also Supplementary Figure 1). The targets were never presented directly in the fMRI task but, instead, the neural representations of each target object were probed using six different sets of clue triplets. Over all trials, the participants guessed the implied identities of the target objects at a high accuracy (mean = 93.3%, SD = 3.1%, min = 87.3%, max = 98.0%; see also Supplementary Table 1).

In the first analysis, we tested whether we can use the corpus-derived semantic coordinates of the target objects to decode the BOLD activation patterns elicited using clues. In order to optimize the decoding accuracy, we averaged the BOLD activation maps of the six trials for each target object. Furthermore, we restricted the analyses to a subset of voxels ($n = 500$) that showed a consistent activation pattern across the six trials (i.e., stability selection; cf. [32,33]). The measurement and analysis protocol for the machine learning analyses is detailed in Fig. 1.

The overall level of classification accuracy using the semantic coordinates of the unpresented target objects was high, ranging

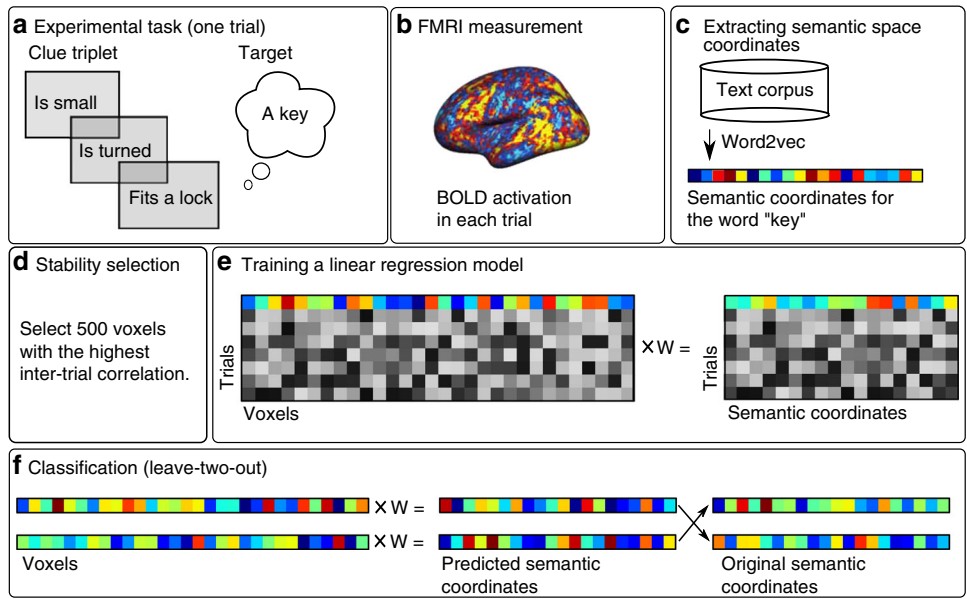

**Fig. 1** Measurement and analysis protocol. **a** Each trial in the experimental task consisted of a clue triplet representing a given target object, e.g., "key". **b** The BOLD activation during each trial was measured. **c** Word2vec model was used to extract semantic space coordinates for the target objects (using either the target word or the feature words). **d** In the stability selection stage, the voxels showing the highest consistency in activation patterns across trials were selected for the analysis. Stability selection was not applied in the single-trial analysis. **e** A training set (without two test-items) and linear regression were used to map the activation patterns of each target object to its respective semantic coordinates. **f** The resulting mapping was evaluated by using it to predict the semantic coordinates of the two left-out objects based on BOLD activity. This scheme was repeated for all possible leave-two-out pairs. FMRI functional magnetic resonance imaging, W regression weights

from 76.1% to 93.3% correct classifications across subjects (mean = 87.2%, SD = 4.9; Fig. 2, see also Supplementary Table 1). The null-distribution for chance level performance was determined through a permutation test in which the relationship of each feature vector and its target was randomized in the training set. This process was iterated 1000 times, each time using a randomly selected participant's brain activation data. Based on the resulting distribution, decoding accuracies > 61.5% were deemed significantly better than chance ($p < 0.05$); this threshold was exceeded by a comfortable margin for all subjects.

Decoding across semantic category (e.g., elephant vs. car) was expectedly more accurate (mean = 94.2%, SD = 5.0%, min = 81.1%, max = 99.3%; permutation test: $p < 0.05$) than decoding within a semantic category (e.g., elephant vs. giraffe; mean = 64.5%, SD = 7.1%, min = 49.8%, max = 74.0%; permutation test: $p < 0.05$). Decoding accuracies across semantic category were significant (permutation test: $p < 0.05$) in all participants, whereas decoding accuracies within a semantic category were significant in 12 out of 17 participants (see also Supplementary Figure 2 for a confusion matrix).

The aforementioned analysis yielded a bi-directional mapping between BOLD activation patterns and the corpus-derived semantic space. This mapping can be used to predict BOLD activation patterns to any number of novel objects in the text corpus based on their semantic space coordinates. In the Supplementary online material (https://aaltoimaginglanguage.github.io/guess/), the recorded BOLD activation patterns for each target object are visualized along with their semantic space coordinates. Furthermore, we included additional targets to demonstrate how the mapping can be used to predict the BOLD activation patterns for novel targets.

**Best decoding is achieved using a rich set of item features**. We next examined whether we can demonstrate pattern completion

in the BOLD activity patterns. Specifically, we tested whether the neural representations elicited by the clues are better defined through semantic coordinates obtained as a summation of all available features linked to a given target object, as compared to only using the exact clues presented to the participants. For this, we used a single-trial model where no averaging was performed across the six repetitions of the same target concept and all voxels in the grey matter were used (i.e. no stability selection). We determined the null-distribution of chance level performance for the single-trial analyses in the same way as for the analyses on averaged data. Based on the resulting distribution, decoding accuracies > 53.6% were deemed significantly better than chance ($p < 0.05$).

The brain activation patterns for each trial were predicted using semantic coordinates obtained via different models: (1) the last clue of each triplet ("Clue 3"), (2) sum of the three clues of the triplet ("Clue 1 + 2 + 3"), (3) the target object and (4) sum of the full list of semantic features typically associated with the target object ("All available features"). The best approximation for the multitude of semantic features associated with each target object was obtained by using a list of behaviorally produced object features from the Centre for Speech Language and the Brain dataset (henceforth, the CSLB features[25]). In addition, we generated two models which excluded those clues that were explicitly provided to probe the target concept. In one variant (5), we mixed the clue sets across blocks such that the semantic coordinates of a given trial were constructed using the clue features of another trial with the same target object ("Mixed clues"). This way, the clues used to predict the brain activation patterns are not the same as those actually presented to the participant (e.g. for a trial where we would probe elephant using clues "has legs", "is thick-skinned" and "has a long trunk", we would decode the brain activation patterns using features "gray", "herd", "tusk", i.e. clues from another block). In the last model (6), we included all features from the CSLB norm data that were

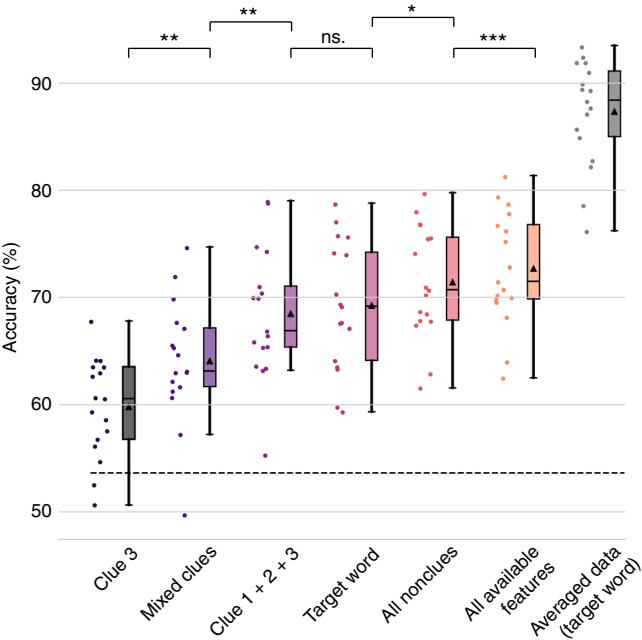

**Fig. 2** Decoding performance using different models. For each model, the raw data points are indicated as a jittered scatter plot on the left side of the summary boxplot and in the same color. Here, each dot represents the decoding accuracy across all pairwise-classifications ($n = 64,620$) for one participant using a given model. Central tendencies for each model are illustrated as a boxplot (the band indicates the median, the box indicates the first and third quartiles and the whiskers indicate ±1.5 × interquartile range). The black triangle within each box indicates the mean decoding accuracy across participants. The level of significant decoding performance ($p < 0.05$, based on a permutation test) is marked as a dashed line. The significance of the pairwise two-sided $t$-tests (Bonferroni-corrected for the number of pairwise comparisons performed) between the mean accuracy scores of the different models is indicated on top (*$p < 0.05$, **$p < 0.01$, ***$p < 0.001$, ns = nonsignificant)

not presented as clues ("All nonclues"; akin to the "All available features" model above, but excluding the nine clue features used in the guessing game task). For all these models, we established a semantic coordinate using the word2vec model and applied the same procedure as for the target words (Fig. 2).

The best performing model was the "All available features" model, i.e. the one where the resulting semantic coordinates incorporated the combination of all CSLB features of a target object (see also Supplementary Figure 3 for a confusion matrix). This model contains numerous features for a given object: both those included in any one of the trials, as well as those never presented to the participant during the course of the entire experiment. For this model, decoding across semantic category was significant in all 17 participants (mean = 77.2%, SD = 6.3%, min = 64.4%, max = 87.3%). Decoding within semantic categories was significant in 16 out of 17 participants (mean = 58.1%, SD = 3.0%, min = 52.6%, max = 63.9%). Within an individual semantic category, the highest decoding accuracy was achieved for tools (mean = 61.7%, SD = 6.0%, min = 49.8%, max = 72.5%; significant decoding accuracy in 16/17 participants), followed by vehicles (mean = 60.8%, SD = 5.1%, min = 51.4%, max = 72.4%; significant decoding accuracy in 15/17), animals (mean = 56.2%, SD = 3.4%, min = 50.7%, max = 65.7%; significant decoding accuracy in 14/17 participants) and fruits and vegetables (mean = 53.7%, SD = 4.3%, min = 46.2%, max = 61.7%; significant decoding accuracy in 9/17 participants). These results suggest

that the semantic coordinates were sufficiently detailed to distinguish items within the same semantic category.

In the next step, we attempted to directly test whether the task elicited features that were not explicitly presented (i.e. whether pattern completion took place). We focused on the two models that included other features of the target objects than the ones that were explicitly provided in a specific trial (Fig. 2; i.e., Mixed clues (model 5) and "All nonclue features" (model 6)). In the "Mixed clues model", there was no overlap between the clue features presented to the participant and those used to create the word2vec semantic coordinates. Using this model, the overall decoding accuracy was significant for all but one participant (see Fig. 2). When the model combined all available features from the CSLB dataset excluding the nine clues used in the task, a significant decoding accuracy was obtained for all 17 participants (see Fig. 2). These results indicate that we can reach a significant decoding accuracy even when the explicitly presented clues are excluded from the model.

The "All available features" model achieved the highest decoding accuracy out of the six models tested. The next best model was the All nonclues model (All available features vs. "All nonclue features", two-sided $t$-test: $t(16) = -9.0$, $p < 0.001$) followed by the Target word model, on par with the Clue $1 + 2 + 3$ model ("All nonclue features" vs. Target word, two-sided $t$-test: $t(16) = -3.6$, $p = 0.1$; Target word vs. Clue $1 + 2 + 3$, two-sided $t$-test: $t(16) = 1.0$, $p = 1.0$). The next best model was the Mixed clues model (Mixed clues vs. Clue $1 + 2 + 3$, two-sided $t$-test: $t(16) = -4.5$, $p = 0.002$). The model using only the last clue of each triplet performed at the lowest level of all (Mixed clues vs. Clue 3, two-sided $t$-test: $t(16) = -4.2$, $p = 0.003$). The summary measures and distributions of the decoding results of all models are presented in Fig. 2.

**Item identity can be decoded using PRC activity**. We conducted an additional region-of-interest (ROI) decoding analysis where we restricted the analysis to the bilateral PRC, given our a priori hypothesis regarding the importance of the PRC in combining the features together into object representations. For this ROI analysis, we averaged the BOLD signal in the bilateral PRC across the six repetitions of the same target object and used the semantic coordinates constructed from all available features. This analysis resulted in a statistically significant decoding accuracy in 15 out of 17 participants (mean accuracy = 69.1%, SD = 6.0%, min = 59.5%, max = 79.7%). This result indicates that the meaning of the target objects can be decoded based on the BOLD signal in the PRC alone. The result further supports the hypothesis that the PRC is involved in decoding object meanings that are reconstructed from a limited set of clues.

**RSA shows regions where BOLD reflects semantic similarity**. We used a single-trial searchlight RSA to determine the brain-wide set of regions where BOLD activation patterns reflected the semantic similarities of the implied target objects. Here, separately for each searchlight sphere, a representational dissimilarity matrix (RDM) was computed based on the BOLD activation patterns[46]. This RDM reflected the distance (1–Pearson's correlation) between activation patterns for each pair of trials. We tested for significant correlations between these activation-pattern RDMs and a model RDM which was based on the All available features model, i.e. the best performing model from the zero-shot decoding analysis. Here, the semantic coordinate of each trial was calculated as the sum of semantic coordinates of All available features of the implied target object. The model RDM was computed using the cosine distances between these semantic coordinates.

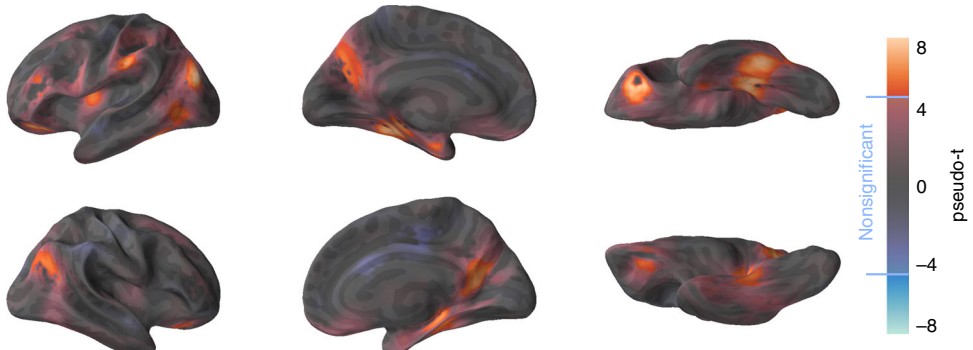

**Fig. 3** Single-trial RSA searchlight results. Brain regions whose activation patterns correlated with the semantic similarity among the target objects when each object was represented by a combination of its full semantic feature set. Pseudo-*t* values for each voxel are projected onto the surface of a template brain. The vertices where the pseudo-*t* value was not statistically significant are semi-transparent (significance level set by a permutation test at FWE-corrected level of $p < 0.05$, pseudo-$t > 4.82$). The uncorrected *t*-maps rendered on 3D volumes are provided online (https://aaltoimaginglanguage.github.io/guess/)

| Table 1 Peaks of the significant single-trial RSA searchlight clusters | | | | | | |
| --- | --- | --- | --- | --- | --- | --- |
| **Anatomical location** | **Pseudo-t** | **Cluster extent** | **Voxel-level p(FWE)** | **x** | **y** | **z** |
| Left middle occipital | 8.1 | 591 | <0.001 | −41 | −78 | 21 |
| Left inferior frontal | 8.0 | 111 | <0.001 | −28 | 35 | −13 |
| Left inferior temporal | 7.9 | 465 | <0.001 | −41 | −40 | −16 |
| Left supramarginal | 6.8 | 76 | <0.001 | −59 | −28 | 31 |
| Right parahippocampal | 6.5 | 139 | 0.002 | 31 | −34 | −10 |
| Left inferior frontal | 6.3 | 58 | 0.002 | −50 | 32 | 18 |
| Left insula | 6.3 | 64 | 0.002 | −34 | −6 | 9 |
| Right middle occipital | 6.1 | 118 | 0.003 | 47 | −68 | 27 |
| Left inferior parietal | 6.0 | 44 | 0.004 | −41 | −40 | 43 |
| Left PRC[a] | 5.7 | 33 | 0.006 | −28 | −6 | −31 |
| Right superior frontal | 5.7 | 36 | 0.006 | 22 | 32 | −16 |
| Right PRC[a] | 4.9 | 1 | 0.046 | 34 | −9 | −31 |

The results are based on a permutation test (10,000 iterations) and family-wise-error (FWE) – corrected $p < 0.05$. Anatomical location is based on the AAL atlas[62] unless indicated otherwise
[a]Based on anatomical localization of the PRC in refs. [47,48,70]

The RSA resulted in 12 significant clusters (family-wise error corrected (FWE) $p < 0.05$ based on a permutation test; Fig. 3). The clusters comprised bilateral ventral-stream regions (inferior temporal gyrus and parahippocampal gyrus) including the bilateral PRCs (Table 1, see e.g. refs. [47,48]). Other clusters were centered in the bilateral middle occipital cortices, left inferior frontal gyrus (two peaks), left supramarginal gyrus, right superior frontal gyrus and left insula. The locations of all peak voxels are reported in Table 1. Analyses on the alternative models used in the decoding analyses (i.e. Clue 3, Clue 1 + 2 + 3, Target, Mixed clues, All nonclues) resulted in largely similar anatomical patterns. These results are provided in the associated online material (https://aaltoimaginglanguage.github.io/guess/).

## Discussion

Humans are able to recognize objects and understand their rich meanings even when only limited information about them is available. In this study, we simulated such a situation by presenting the participants with brief verbal descriptions of 60 objects and asking them to guess the identity of each of them. We showed that it was possible to decode the implied target object with high accuracy without ever showing the object explicitly, suggesting that the clues triggered a coherent representation of the target object. The single-trial results further demonstrated that the brain activation patterns elicited by the guessing game paradigm indeed contained more information about each target

object than what was initially given as input in the experiment. This suggests that the entire neural representation of an object became activated based on partial stimulation in the form of only few features. This finding provides neuroimaging evidence on semantic pattern completion whereby limited information in the environment is used to reconstruct coherent object representations.

Distributed accounts of semantic representations postulate that neural representations of objects can be modeled as unique and consistent distributions of activity across a set of perceptual and semantic feature nodes (see e.g. refs. [5,6,49]). This model has been successful in describing, not only the healthy semantic system[11,26], but also patterns of semantic impairments associated with brain damage[9,42,50]. Importantly, such a feature-based distributed system also gives an account on how information is reconstructed from incomplete patterns of information. Specifically, the activation in a subset of feature nodes is postulated to propagate in the network based on connection weights which, in turn, are based on experience on co-occurence[8,11,51].

We found evidence of semantic pattern completion by combining neuroimaging and machine learning with corpus-derived coordinates of objects and their features in a shared semantic space. Using single trials, we found the best mapping using semantic coordinates created by summing many features for each given object, including features never presented to the participant. This model performed significantly better than the model using the semantic coordinates for the target object alone or that using

the sum of the clues presented to the participant in a given trial. This finding therefore provides neuroimaging markers of pattern completion, that is, that activation in a subset of object features leads to activation in a network of features associated with a given object entity[8,9,11,51].

Further evidence for semantic pattern completion was offered by models incorporating features that had not been presented to the participant. We were able to decode brain activation patterns using non-overlapping clues from other trials (above chance-level decoding for 16 out of 17 participants). Moreover, a high level of decoding accuracy could also be achieved by using a rich set of target features (akin to the All available features model) while excluding those nine features that were used as clues in the experiment. Indeed, this analysis performed significantly above chance level for all participants and better than the other models, surpassed only by the model that incorporated All available features including the nine features used as clues in the experiment. This means that the guessing game task activated features that were incidental to the task, i.e. those that were not given as input in the experiment. We suggest this finding provides further support for the notion of pattern completion in the current study.

Interestingly, the decoding accuracy was higher for the All available features model, combining word2vec vectors over a rich set of features, than the model using the word2vec vector for the target word only. We speculate that the All available features model performs better than the Target model as it combines several target-appropriate features together, and therefore creates a vector representation which explicitly contains information about the relevant features of the concept. This may be a richer representation than that inferred from simple co-occurrence counts of isolated target words or individual clues in the corpus.

A combination of the present RSA results and previous research on visual processing may provide insights into the neural basis of semantic pattern completion of objects. Studies on visual object recognition suggest that pattern completion in the visual domain occurs in the ventral stream via recurrent connections[20] (see also ref. [35]). The importance of the ventral stream in processing also the meaning of visual objects has been demonstrated by Clarke and Tyler[41]. In that study, the authors presented participants with a large set of naturalistic color photographs and showed that regions in the lateral occipital cortex and ventral stream were sensitive to the semantic similarity of the presented visual objects. The anatomical pattern of the RSA results in the ventral stream in our study bears remarkable similarity to those of Clarke and Tyler[41] despite the fact that we never showed images or pictorial stimuli to our participants. It is possible that reading descriptions about objects, such as in the guessing game task, recruits embodied visual representations of objects and, thus, recruit the ventral stream system (see also ref. [52]). Our results suggest that this system may also play a role in pattern completion of meaningful object representations.

The significant link between brain activation patterns and the similarity of semantic features of the objects was observed at a high level of ventral stream hierarchy. A ROI analysis also demonstrated that we could decode the identities of objects based on the BOLD response from PRC alone (significant decoding accuracy: 15/17 participants). Similarly, in the RSA analysis, the clusters showing sensitivity to the semantic similarity of the target objects included the bilateral PRC, located at the apex of this system. Importantly, this region has been highlighted in item-specific processing of object meaning in a visual object naming task[41] and a property-verification task with pictures and words[52]. Our findings extend these results in showing that this region is involved in item-level processing of objects even in the absence of pictorial stimuli. Moreover, the findings corroborate those by Taylor and colleagues[38,39] who showed that the PRC is involved

in binding features from multiple modalities. Importantly, the current study demonstrates that these features need not be visual or auditory but they may also come in the form of more abstract semantic properties. Therefore, this study strongly supports the claim that this region is involved in processing object meaning.

We found a set of other regions that were associated with the semantic similarity of the target objects in addition to those in the ventral stream. These regions include the temporo-parietal junction and inferior frontal cortex, whose involvement may reflect the verbal nature of the task and conceptual and lexical preparation for the verbal response[53]. Other regions include the bilateral retrosplenial cortex, which in previous research has been associated with visual imagery and memory, and whose involvement can partly be explained by specific strategies employed in the task (for a review, see ref. [54]). Importantly, the network of areas revealed by this analysis are also likely to be a reflection of the distributed nature of the semantic representations themselves. Indeed, Huth and colleagues[30,31] showed that semantic information in the brain is organized systematically as smooth gradients reflecting semantic similarity in wide-spread and distributed regions of the brain. Therefore, we postulate that the regions highlighted by the RSA searchlight analysis are relevant in representing concrete objects such as those targeted in the current experiment.

We note that the RSAs yielded largely overlapping anatomical patterns for all models tested in this study (i.e. Clue 3, Mixed clues, Clue $1 + 2 + 3$, Target word, All nonclues, All available features; see https://aaltoimaginglanguage.github.io/guess/). Thus, despite the fact that we found significant differences in decoding accuracy in the zero-shot decoding analysis, all six models had similar anatomical patterns of correlation with BOLD activity which differed primarily in extent. Indeed, we would like to emphasize the fact that each one of the six models contained relatively high-level conceptual information about objects (i.e. as compared to e.g. low-level perceptual features). We propose that this explains the overlapping patterns of activity in regions associated with semantic object processing.

The selection of a statistical threshold is to some extent arbitrary. As illustrated in Fig. 3, there were multiple regions where correlations failed to reach the selected statistical threshold, but which may nonetheless be functionally linked with item-level semantic processing of objects. The full range of effects can be seen in the uncorrected statistical images which we have made available https://aaltoimaginglanguage.github.io/guess/). These data demonstrate that effects which may seem discrete with the chosen threshold in fact reflect a continuum of effects over a larger region.

We postulate that the neural basis of semantic pattern completion may differ from that of episodic memories, which has been attributed to the hippocampus[1–3,15–17,19,55,56]. Specifically, we suggest that the pattern completion of semantic information about objects partly takes place in the ventral stream and the PRC. We suggest that semantic pattern completion may require the disambiguation of feature combinations (i.e. the clues in the present task) and, therefore, rely on the complex representations of feature conjunctions provided by the PRC[37,57]. Thus, pattern completion of semantic object memories may take place in the same system that is involved in processing and perceiving objects. It should be noted that similar effects have been demonstrated for mental imagery (see e.g. ref. [58]) and in the current framework we cannot dissociate the process of pattern completion from the end result of mental imagery (i.e. the target concept). Therefore, further research is required to provide a mechanistic account of the process of semantic pattern completion in the brain.

Pattern completion of semantic information is a frequent phenomenon in our everyday life. The brain automatically and

effortlessly takes advantage of clues in our environment with prior knowledge about the meaning of objects we encounter. Otherwise, we would not be able to make sense of the world around us. The present study aimed to find empirical neuroimaging evidence for this process. Indeed, the present results demonstrate that we can use clues to elicit a representation that contains more information about the object's meaning than what was provided as input. We do this by showing that (1) the highest decoding accuracy was achieved by combining the rich set of features associated with each object and that (2) we could decode features incidental to the task, i.e. features that were not presented. We suggest that these results reflect the spread of activation in the neural networks as suggested by computational accounts on semantics[1–3,56]. It should be noted, however, that although we deduce that this is most likely the case, we cannot directly observe the pattern completion process with the current design owing to the sluggish haemodynamic response. Future studies are needed to demonstrate the actual process of semantic pattern completion in the brain. We believe that the present methodology (guessing game, word2vec and zero-shot decoding) combined with more time-sensitive brain imaging techniques will be a very fruitful approach in understanding the neural dynamics of that process.

Using stability selection and data averaged across all six trials of the same target object resulted in a high decoding accuracy that was comparable to those in studies which have used colored photographs as stimuli[33]. In the past, semantic category-level decoding performance has, at least partly, been attributed to a robust response to low-level visual features[59]. However, the present results demonstrate that these visual attributes are not necessarily needed in order to achieve high-level decoding accuracy. Moreover, we suggest that the guessing game paradigm used in this study is highly engaging from the participant's point of view, leading to elaborate processing of the target stimuli. Therefore, we suggest that it is particularly well-suited for experimentally accessing semantic representations.

The present neuroimaging study used a novel experimental design to demonstrate that the brain completes patterns of fragmented information into meaningful, coherent semantic representations. This design, coupled with our machine learning models, allowed us to show, for the first time using neuroimaging, that the brain takes advantage of very limited information and enriches it with prior knowledge of object meaning. The present results give strong support for the distributed, feature-based models of semantics in the brain and suggest that rich representations of object meanings are partly supported by the ventral stream and the PRC.

## Methods

**Participants**. Eighteen native Finnish-speaking, right-handed individuals with no history of developmental or acquired language or other neurological disorders participated in the study. The participants were recruited through student mailing lists in the Aalto University. One participant chose not to finish all measurement runs and was therefore excluded from data analysis. Thus, the sample consisted of 17 individuals (mean age = 20.9 years; SD = 3.3 years, min = 18 years, max = 31 years; mean education = 12.4 years, SD = 1.5 years, min = 12 years, max = 18 years; 10 identified themselves as females, and seven as males). All of the participants gave a written informed consent before participating in this study. The study was approved by the Aalto University Research Ethics Committee.

**Stimuli**. The stimuli consisted of 540 brief verbal descriptions of 60 target objects in Finnish (9–29 characters including spaces, mean = 17.5, SD = 3.6). Fifty-eight target objects were selected from the CSLB property data set[25]. We additionally included two target objects that were not part of the CSLB data [forklift (Finnish: 'trukki'), and metro (subway) (Finnish: 'metro')]. One fourth (n = 15) of the target objects fell into each of the following semantic categories: animal, fruit/vegetable, tool and vehicle. We created nine clues (i.e., descriptions) per each target object by translating and adapting semantic features from the CSLB data. For the two objects not included in the CSLB data set, we selected six features from that set that applied

to the target object and additionally created three new highly distinctive features. We also created 29 new clues (5.3% of clues in total) in cases where sufficiently many suitable clues were not available in the CSLB data set. The first, second and third clues were matched on length across the four semantic categories (pairwise t-test: p > 0.59 for all).

The nine clues assigned to each target object were further divided into three clue triplets. When feasible, the presentation order of the clues within a triplet was sorted such that the first clue in each triplet was the least distinctive (e.g., 'has four legs'), and the following two clues increasingly distinctive (e.g., 'is found in the savannah' > 'has a trunk') based on the CSLB feature norm data[25]. The purpose of this approach was to ensure that the participants would guess the target object approximately at the same stage (i.e., at the third clue).

Each individual clue was repeated at least twice in the fMRI experiment, once in Set 1 and once in Set 2, with the two sets presented on different days. The clue combinations were rearranged such that each clue's position in a triplet was retained, i.e., the first clue of the triplet in Set 1 was always the first clue of a triplet in Set 2, but the other clues it was grouped with were not identical in both sets. This procedure resulted in six unique clue triplets for each target object which were presented in six separate blocks. The order of sets (across measurement days) and blocks (within a set) was balanced across subjects. The full list of clues (Set 1) can be found in https://aaltoimaginglanguage.github.io/guess/.

**Procedure**. The fMRI experiment was conducted on 2 days, with three measurement sessions (i.e., blocks) on each day. We divided the data acquisition into two separate days to ensure that the participants would be able to sustain attention throughout the experiment. The two measurement days were on average 10 days apart (mean = 9.9, SD = 7.9, min = 6 days, max = 35 days) with each fMRI measurement lasting ca. 45 min in total. Each trial started with a fixation cross ('+', duration: 300 ms) after which the clues were presented one after another. The clue duration was 1000 ms and the first two clues were followed by a blank screen for 200 ms. The third clue was followed by a jittered interval (mean = 8.0 s, min = 4.0 s, max = 11.8 s), after which a string of hash characters '################' was presented for 1000 ms, prompting the participant to overtly name the target object (Fig. 4). The interval between the final clue and the naming prompt was relatively long as we attempted to minimize the overlap between the peaks of the BOLD signals. The naming condition was followed by a jittered interval (mean = 4.0 s, min = 2.3 s, max = 6.2 s) after which the next trial started. The jittering was generated using efMRI version 9 (Chris Rorden, Columbia, SC, USA, www.mricro. com). The black text stimuli were presented on a gray background. There were two 18 s rest periods in each measurement session. The rest trials were signaled by a pair of hyphens '--' that the participant was asked to fixate while remaining still.

**Functional MRI data acquisition**. Participants were scanned with a Siemens 3 T Skyra Magnetom MRI device using a custom 30-channel receiver head-coil. We acquired echo-planar imaging (EPI) volumes in axial oblique angle using an acquisition matrix of 64 × 64 with 3.1 mm × 3.1 mm × 3.1 mm voxel dimensions. The following acquisition parameters were used: TE = 32 ms, TR = 2.4 s, flip angle = 90°, slices = 41, FOV = 200 mm, phase resolution = 100%. A structural T1-weighted MPRAGE volume was also acquired (TE = 3.3 ms, TR = 1.1 s, slices = 176, FOV = 256 mm, phase resolution = 100%).

The stimuli were controlled using Presentation® 15.0 software (www.neurobs. com) running on a Dell Optiplex 960 PC. The stimuli were projected to a mirror mounted on the head-coil using a Panasonic PT-DZ110XEJ projector with 1920 × 1200 resolution and 60 Hz frequency. Participants' verbal responses were recorded using an OptoAcoustics (Or-Yehuda, Israel) FOMRI-III optic microphone with OptoActive noise control. The microphone was mounted on the head-coil.

**Semantic space from text corpus data**. The model of semantic space used in the decoding was estimated from a 1.5 billion token Internet-derived text corpus in lemmatized Finnish[22]. The semantic space was built using a word2vec skip-gram model with a maximum context of 5 + 5 words (5 words before and after the word of interest)[22]. The skip-gram model is a fast and efficient method for learning dense vector representations of words from large amounts of unstructured text data. The objective is to find vector representations that are useful for predicting the surrounding words in a sentence given a target word[23,44]. The code is available online at https://code.google.com/archive/p/word2vec, and the word vector data set used is available online at http://bionlp-www.utu.fi/fin-vector-space-models/fin-word2vec-lemma.bin. The word vectors of the model have the dimensionality of 300, and they were used in the machine learning analyses and the RSA[46]. Note that single dimensions of the semantic space are not interpretable.

Word2vec was used to acquire altogether six sets of semantic space coordinates: (1) the last single clue of the triplet that was used as the onset for the fMRI response (Clue 3); (2) the sum of the first, second and third clue of the triplet that were used to probe a given target word (Clue 1 + 2 + 3), (3) the target word alone (Target word) and (4) the sum of the semantic coordinates of all features for a given target object available in the CSLB data set (All available features)[25], including features that were never presented to the participant. In addition, we generated two models which excluded the clues used to probe the target concept: (5) in one of these models, we mixed the clue sets across blocks such that the semantic coordinates of a given trial

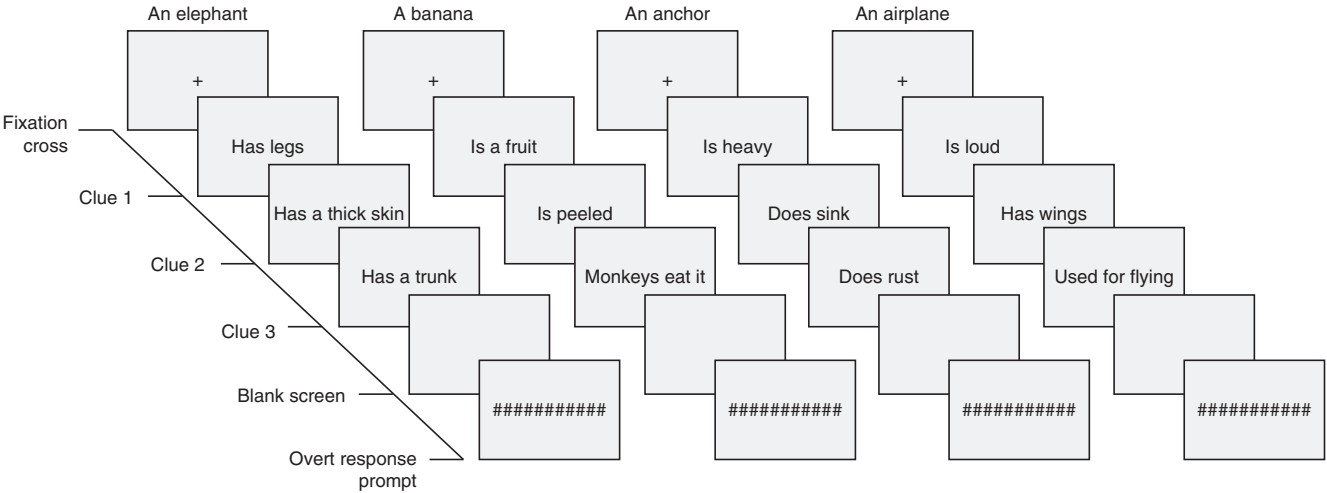

**Fig. 4** Examples of stimuli and experimental design in fMRI. Three clues were shown one at a time, after which the participants were asked to guess which object they describe (e.g., here: an elephant, a banana, an anchor, an airplane). A string of hash characters prompted the participant to utter the name of the target object. The target object itself was never presented to the participants before or during the experiment, either pictorially or as a word, and no feedback regarding correct or incorrect answer was provided

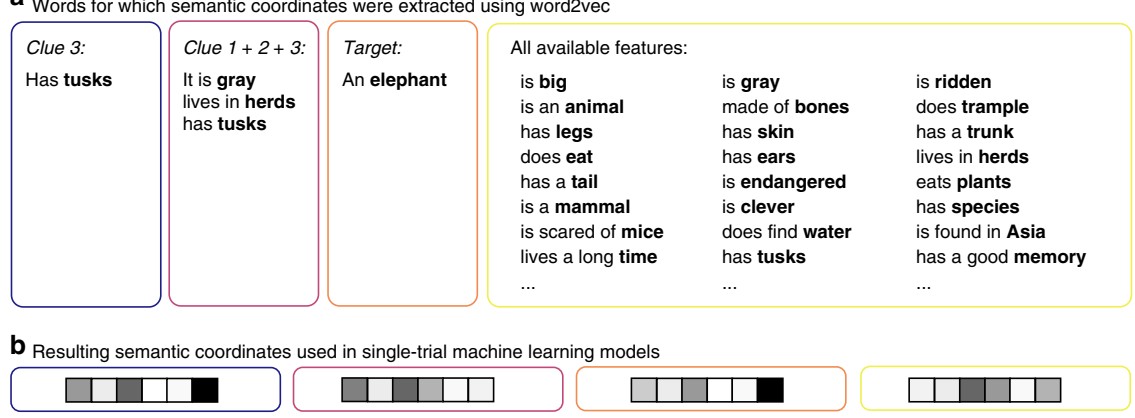

**Fig. 5** Examples on how the different models were constructed. **a** The key word whose semantic coordinates were built using word2vec is shown in boldface. The semantic coordinate was either based on one word (i.e., Clue 3 and Target) or several words (i.e., Clue 1 + 2 + 3 and All available clues) in which case the final semantic coordinate was a sum of the semantic coordinates of all words in the respective model. **b** The sum of the resulting semantic coordinates (i.e. one 300-dimensional vector per each item) was entered into the zero-shot decoding analysis

were constructed using the clue features of another trial with the same target item. This way, the clues used to predict the brain activation patterns were not the same as those that had been presented to the participant (e.g. for a trial where we would probe elephant using clues "has legs", "is thick-skinned" and "has a long trunk", we would decode the brain activation patterns using features "gray", "herd", "tusk", i.e. clues from another block). In the final model (6), we calculated the sum of semantic coordinates of all such features that were not presented in the guessing game task for a given target concept (akin to the All available features model but excluding the nine clue features used in the guessing game task).

The semantic coordinates were built in the following way. For each trial, we used word2vec to extract semantic coordinates for the implied target word, as well as all corresponding CSLB features and the clues used in the task. In cases where the clue/feature consisted of more than one word, we selected and lemmatized one key word (e.g., has legs→leg) and extracted the corresponding semantic coordinate from the corpus. For models, which combined many features together (i.e. Clue 1 + 2 + 3, Mixed clues, All nonclues, All available clues), we extracted the semantic coordinates of all features' key words and then calculated the sum of the resulting semantic coordinates (see Fig. 5). Thus, all models used in the single-trial analyses resulted in a 360-by-300 matrix (i.e., number of trials × number of dimensions of the semantic space). In the analysis with averaged data, we used a 60-by-300 matrix (i.e. number of target objects × number of dimensions in the semantic space).

**FMRI data preprocessing**. The preprocessing was performed using SPM8 software (Wellcome Trust Centre for Neuroimaging, University College

London, UK) running on Matlab (MATLAB 2014a, MathWorks Inc., Natic, MA). The EPI volumes were first corrected for slice timing and head motion and co-registered to the structural volume of the same participant. We used a General Linear Model approach, where the model contained the head motion and session parameters as nuisance regressors as well as high-pass filtering. Each of the target objects in each of the six blocks was modeled by convolving a canonical hemo-dynamic response function from the onset of the last clue of a triplet. All analyses were run on native-space unsmoothed data. For visualization purposes, the data was co-registered to Montreal Neurological Institute (MNI) reference space[60]. Anatomical labeling was based on the AAL atlas[61] unless otherwise cited.

**Zero-shot decoding analyses**. The machine learning analyses were run on Python 3 (www.python.org) using Anaconda3 distribution and the scikit-learn module[62]. The machine learning models implemented in this study evaluated the contributions of the brain activation patterns to each of the 300 dimensions in the semantic space (Fig. 1). The aim of these analyses was to test whether we can establish a statistically significant mapping between the brain activation patterns and the word2vec semantic coordinates.

The models were trained by using a subset ($n = 58$) of the altogether 60 targets and the respective multi-dimensional semantic coordinates such that, in the end, each semantic dimension was associated with a particular weighted activation pattern. For this, we used multiple regression with regularization parameters. The trained model can be used to predict the brain activation patterns of any novel concept outside the training set for which the corpus-derived semantic coordinates are available.

The model was evaluated after the training such that the predicted semantic coordinates of the two left-out objects were compared with the original corpus-derived ('true') semantic coordinates. The classification outcome was determined using cosine distance. We evaluated the level of statistical significance using a permutation test with 1000 iterations, randomly selected subjects and randomly shuffled order of the semantic coordinates across the target objects.

**Zero-shot decoding on averaged data.** In this analysis, the six repetitions with unique clue triplets for a given target object were averaged together into a single BOLD activation map using stability selection as described below. The zero-shot decoding model was trained by using 58 of the target items and the training data was used to predict the semantic coordinates of the two left-out target items. The training and evaluation process was iterated 1770 times to cover all leave-two-out combinations.

We focused the machine learning analysis of averaged data on a specific subset of voxels that showed a consistent activation pattern across the six trials of each target object[32,33]. First, we masked the native space beta images using an individual gray matter mask extracted from the SPM segmentation. We then extracted beta values for each voxel of each repeated trial ($n = 6$) of each object ($n = 58$, i.e., excluding the leave-two-out objects at each iteration). We then calculated pairwise Pearson correlations across the six repetitions of each target object and averaged the correlations over the 58 target objects in the training set. Finally, the 500 most stable voxels, i.e., those with the highest average correlation, were selected for further analyses.

**Single-trial zero-shot decoding.** In the single-trial analysis, no averaging was performed over the six trials of the same target object, but each trial using a unique clue triplet was considered a separate item. The brain activation patterns related to each trial were then used to predict the semantic coordinates (for details, please see section: Semantic space from text corpus data). First, the test pair was selected after which the other $5 + 5$ trials corresponding to the same target concepts were removed from the training set. Thus, the zero-shot decoding model was trained on 348 trials, i.e. all 12 trials representing the two targets we tried to predict were excluded from the training set. Note that we did not use stability selection in the single-trial analysis, since there were no repeated trials over which stability selection could sensibly have been performed. Furthermore, as each trial had a different set of clues, we did not want to potentially wipe out this variability.

In the last step, we wanted to test which one of the trained models (i.e. using different sets of semantic coordinates as described above) provided the best mapping to the observed brain activation patterns. To this aim, the decoding accuracies between different models were compared using a pairwise t-test using a Bonferroni correction.

**Visualization of the zero-shot results.** To demonstrate the mapping between the brain and semantic space learned by the zero-shot decoding algorithm, we have created an interactive visualization (https://aaltoimaginglanguage.github.io/guess/) that shows, for each target object, its coordinates in the semantic space and the corresponding BOLD activation pattern, averaged across the six trials. T-distributed stochastic neighbor embedding (t-SNE)[63] was used to obtain a two-dimensional visualization of the semantic space, and pycortex[64] was used to visualize the BOLD activation pattern. To illustrate that the mapping between the brain and semantic space is defined at all coordinates, we added 19 new targets (mouse, parrot, chicken, goat, lynx, peach, grapefruit, beetroot, broccoli, lettuce, plane, screw, plate, watch, tape, tram, tank, dinghy, gondola) to the interactive visualization. By reversing the mapping to obtain a linear transformation between the semantic space and the brain[65], BOLD activation patterns were predicted for these novel items.

**Representational similarity analysis.** In the RSA analysis, we used the single-trial data to maximize comparability with the decoding results. We used searchlight mapping[46] and RSA toolbox[66] running on MATLAB 2014a (The MathWorks, Inc., Natick, Massachusetts, United States) to find regions where similarity of activation patterns (activation pattern RDMs) was related to the semantic similarities of the implied target objects (model RDM).

The model RDM was based on All available features model. That is, the semantic coordinate of each trial was calculated as the sum of semantic coordinates of All available features of the implied target object (see model 4 above). The resulting model RDM was a $360 \times 360$ matrix, where the value in each cell reflects the cosine distance between the semantic coordinates of a pair of trials. The model RDM was compared to activation pattern RDMs which were constructed for each spherical searchlight (radius = 7 mm) across each voxel in the gray matter volume. The activation pattern RDMs were symmetrical $360 \times 360$ matrices, where the value in each cell reflects the dissimilarity (1−Pearson's correlation) of activation patterns between a pair of trials. A whole-brain correlation map was produced by calculating Spearman's rank correlations between the activity-pattern RDMs and semantic model RDMs. The correlation maps were Fisher transformed in order to make them normally distributed and projected back onto each searchlight's center voxel.

The correlation maps of each participant were transformed into MNI space and smoothed at six FWHM. The resulting normalized and smoothed images of each participant were subjected to a group-level statistical nonparametric mapping analysis (one sample t-test) using variance smoothing of six FWHM and 10,000 permutations (SnPM13, version 13.1.06; http://go.warwick.ac.uk/tenichols/snpm). FMRI analyses are prone to increased risk of false positives as statistical tests are performed on a very large number of voxels. To deal with this problem, we indicate the pseudo-t values that survive the voxel-level FWE corrected p-threshold < 0.05 (height threshold: pseudo-$t = 4.82$). The uncorrected pseudo-t maps of the main RSA analysis[67] are provided in an online repository: https://aaltoimaginglanguage.github.io/guess/. We further provide the results of alternative RSA analyses, using the remaining models applied in the single-trial decoding analyses.

**Region of interest analysis.** The PRC ROI was based on FreeSurfer's (https://surfer.nmr.mgh.harvard.edu) probabilistic PRC label[68]. This label encompasses the medial bank of the collateral sulcus which corresponds to the Brodmann's cytoarchitectonic field 35, i.e. the transentorhinal cortex[69] (see also refs. [47,48]). The surface-based labels were converted to volume-based ROIs after which the resulting ROI was manually inspected. When necessary, the ROIs were corrected manually such that they continuously covered the entire medial bank of the collateral sulcus.

The BOLD activation maps inside the PRC ROIs were averaged across the six repetitions of each target object. The voxel-wise BOLD signals in the left and right hemispheres were then concatenated resulting in a matrix with 60 rows (number of target items) and $n$ columns, where the $n$ corresponds to the total number of voxels in the left and right PRC ROI. These data were subjected to the zero-shot decoding scheme and used to predict the semantic coordinates of the 60 items. The semantic coordinates were built using the All available features model (i.e. All available features in the CSLB norm data[25]).

**Code availability.** The used software and algorithms are detailed in Supplementary Table 2. The custom code related to the zero-shot-learning algorithm and visualization code is available https://aaltoimaginglanguage.github.io/guess/.

## Data availability
Information related to data availability is detailed in Supplementary Table 2. The relevant data used in this study are available upon request to the editors and reviewers and researchers who meet the criteria for access to confidential data. Qualified researchers may contact the secretary of the Aalto University Research Ethics Committee, Jari Söderström (jari.soderstrom@aalto.fi). Ethical restrictions prevent the authors from making the raw MRI data publicly available, as this would compromise research participants privacy and consent. These restrictions have been imposed by the Aalto University Research Ethics Committee, in compliance with Finnish legislation on Data Protection.

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

## Acknowledgements

We acknowledge the computational resources provided by the Aalto Science-IT project and thank Marita Kattelus for measurement assistance. We also thank Filip Ginter and Jenna Kanerva from the Turku BioNLP Group, University of Turku for providing the Finnish corpus-based model and Dr. Emmanuel Stamatakis for his insights on fMRI design. We also thank Hanna Renvall, Jan Kujala, Mia Liljeström and Anni Nora for insights and helpful discussions. This research was funded by the Academy of Finland (grant #286070 to S.L.K., #310988 to M.v.V, #287474 to A.H., #255349, #256459, #283071 and # 315553 to R.S.), the Aalto Brain Center (M.v.V and T.L.-K.), and the Sigrid Jusélius Foundation (to R.S.).

## Author contributions

Conceptualization, S.L.K., A.H., R.S.; Methodology, S.L.K, M.v.V., R.S., A.F., T.L.-K.; Investigation, S.L.K.; Interpretation: S.L.K., M.v.V., A.H., R.S.; Writing—original draft, S.L.K.; Writing—review & editing, S.L.K., M.v.V., A.F., A.H., T.L.-K., R.S.; Funding acquisition, S.L.K., R.S.

## Additional information

**Competing interests:** The authors declare no competing interests.

