## [Peer Review File · Nature Communications]

Reviewers' comments:

Reviewer #1 (Remarks to the Author):

Reconstructing meaning from bits of information

This is an interesting study where the authors aim to decode semantic information using functional neuroimaging based on the BOLD measurements observed in response to 3 cues presented to the subjects. Understanding how semantic information is represented in the brain constitutes a challenging question that remains poorly understood. This study represents a step in the right direction and provides intriguing results suggesting that even spatially and temporally coarse neuroimaging measurements may provide hints of semantic encoding.

Error bars are essential in any report of scientific data. Data for individual participants should also have error bars in Fig. 2. Lots of other numbers are provided in the text without the corresponding measures of variation (e.g. behavior of 93.3%, among many others, won't list them all in here).

It is not clear exactly how the algorithm is trained and tested. Specifically, do the training data and test data correspond to different presentations of identical cue stimuli? Or can the authors decode information across different cues (e.g. train on completely non-overlapping sets of cues from those used in the test set)?

Figure 3 uses a nasty convention in the fMRI literature that is more hurtful than useful. Values below an arbitrary threshold are portrayed as gray. The problem is that these thresholds are completely arbitrary. It is important to show the entire scale of values for every voxel (i.e., no thresholds at all, no gray in the brain rendering, every voxel should have a t value and hence a color).

There is little information on what the objects are, how similar they are, etc. This is important to better understand the leave-two-out performance and how similar those left out objects are to the rest of the objects.

It is remarkable that performance is so high in this task: e.g., 94.2% accuracy for elephants versus cars. This is much better than what others have reported using much better neurophysiological data. Whatever the authors are doing here, their results are impressive.

Reviewer #2 (Remarks to the Author):

Kivisaari and colleagues study the neural underpinnings of our ability to understand a concept from partial information. They used a task where participants receive clues about the identity, before guessing, during fMRI and applied machine-learning and multivariate techniques to probe the neural representations while hearing the clues. Specifically they ask whether the neural representations reflect the clues, or the entirety of the concept the clues relate to. They conclude that much of the ventral visual pathway, including the PRC, reflect the a complete semantic representation rather than only the clues.

The manuscript is clear and well written, and reflects a very nice piece of research. However, I have two prominent issues relating to (1) whether this reflects evidence from semantic pattern completion, and (2) would like to see extra evidence to bolster the claims made.

Major comments

1. To my knowledge, semantic pattern completion is a novel idea, and on the surface is attractive and intuitive. However, I'm not sure it's actually a useful construct. After all, when we see a visual object and activate a semantic representation, not all aspects of semantics will be visually present – so is this also a case of semantic pattern completion? If so, every study on the activation of semantics from images can be seen as evidence for semantic pattern completion. This is not to undermine the idea, as it must certainly be the case we do this, but I'm not sure this study shows us how this happens.

Theoretical issues aside, I feel that both the task, and the GLM used, might also undermine the evidence being presented in favour of semantic pattern completion. In terms of the task, participants know this is a guessing game and receive three clues. The first 2 clues will be relatively more shared (less distinctive) and so people know roughly what kind of thing it'll be. The third clue is the most distinctive, and so at this point they will guess what it is, and so the neural representation may well reflect the guess, and therefore the target semantics. Therefore, it can be argued the neural representations do not reflect the clues alone, but the target itself. The data presented backs this up, but doesn't this more reveal that the task elicited semantic representations rather than feature/clue representations, rather than telling us how these pieces were used to elicit semantic representations.

A related issue is with the GLM used, that is based only on the third clue, and so will reflect the guess (given the above). I feel the process of going through clue 1- 3 would be most revealing in terms of how information is combined.

I feel all these issues make it difficult to make claims relating to semantic pattern completion processes (or the how), rather than an analysis of the result of semantic pattern completion (i.e. a complete semantic representation).

2. I think the data analysis presented is clear and appears robust, however I felt some additional analyses are warranted. In terms of the decoding analysis, it would be useful to know the accuracies for within and between category decoding (or, how specific, semantically, are these predictions). If it is truly a complete semantic representation, rather than at a superordinate category level, then you should see significant within-category effects. Further, given the strong hypothesis about the role of the PRC, something like an additional decoding analysis restricted to the PRC would be very compelling.

I also thought the RSA searchlight analyses were very brief. The only analysis presented is based on the target items, but couldn't this be supplemented with either (a) a comparison between RSA of the target words vs. all available features, or (b) a single trial RSA analysis to match the conditions presented in Figure 2.

Overall, I feel the decoding analysis is aimed at telling us what the nature of the distributed representations are, and the RSA analysis to add spatial specificity. But because the two present different analyses they are a little tricky to combine. I think either expanding the decoding or the RSA would help do this.

3. A final issue is what does it tell us that the word2vec vectors for the target do a worse job than the combined vectors for all the target's features? Is it that the semantics of the targets are not well estimated by word co-occurrence, but they are by the combined word co-occurrence structure given by their features?

Minor issues

Page 11. It wasn't clear to me whether in the RSA analysis the RDM was defined based on word2vec or the target words, or all available features, or just the cosine distance based on feature overlap?

Figure 3 legend. It would be useful to add thresholds, statistical corrections etc for clarity.

I wondered if the authors had considered analyses comparing activation during clue 1/2/3 vs the guess, to see how clue representations incrementally reflect the guess (this may get closer to the how question).

Was there a reason the study was conducted over two sessions?

Reviewer #3 (Remarks to the Author):

Kivisaari et al. present a single fMRI study focussing on “semantic pattern completion” – the retrieval of all semantic features of an object when cued with a subset of object features. They use pattern analysis techniques (pattern classification and RSA) to provide evidence that large scale brain networks show patterns that more closely correlate with those predicting by a model combining all known features of an object, relative to those coding for a subset of cued features.

The work is very interesting, the approach is novel, and the results are compelling. I found the paper relatively easy to read (though see comment 2 below), and the conclusions seem justified. I do have several comments that the authors may find useful:

1. The authors mention pattern completion in relation to episodic memory, citing two largely theoretical papers. There is a very rich literature on hippocampal pattern completion (and neocortical reinstatement), which starts with Marr. Connections with this literature would broaden the scope of the current manuscript, as well as provide a richer empirical and computational background to the study:

Theoretical work:

Marr (1971) Simple memory: a theory of archicortex

Gardner-Medwin (1976) The recall of events through the learning of associations between their parts

McClelland et al (1995) Why there are complementary learning systems in the hippocampus and neocortex

Behavioural work:

Horner & Burges (2014) Pattern completion in multielement event engrams

Vieweg et al. (BioArXiv) Memory Image Completion (MIC): Establishing a task to behaviourally assess pattern completion in humans

fMRI work:

Wheeler et al. (2000) Memory's echo: vivid remembering reactivates sensory-specific cortex

Woodruff et al. (2005) Content-specificity of the neural correlates of recollection

Staresina et al. (2012) Episodic reinstatement in the medial temporal lobe

Horner et al. (2015) Evidence for holistic episodic recollection via hippocampal pattern completion

Review papers:

Hunsaker & Kesner (2013) The operation of pattern separation and pattern completion processes associated with different attributes or domains of memory

Horner & Doeller (2017) Plasticity of hippocampal memories in humans

2. I found the “zero-shot learning and model evaluation” in the methods difficult to follow. Perhaps clearer statements regarding what the training data are and what the test data are, as well as what this analysis is designed to provide evidence for, would help the reader.

3. The analysis comparing the different models provides evidence that the “all available features” model gives the highest classification accuracy. To what extent might this result be driven by the increased number of features, rather than true pattern completion? For example, if presented with cues for an elephant, on one trial I might bring to mind that they are endangered (non-cued) and on another that they have big ears (non-cued). The “all available features” model would work best here, but it doesn’t mean I’m always pattern completing. Instead, I’m bringing to mind different non-cued features on each trial. I wonder whether a model selecting 3 (random) non-cued features could help here. At the least, it would control for the overall number of features (relative to the “clue 1+2+3” model), and would also show you can classify solely on the basis on non-cued features.

4. The RSA analysis only uses the “all available features” model. Did the authors also perform the analyses for the other models? Do they produce largely similar (but somewhat weaker) results, or are there any potentially interesting dissociations? For example, I might imagine that the “target” model might work well further up the ventral visual stream (e.g., PRC) but not in more posterior ventral visual regions.

Re. RESPONSE TO REVIEWERS

Editor's comment: We would expect to see the additional data analyses recommended by Reviewers #2 and #3.

Response to Editor. We have made several modifications to the text and carefully addressed each of the Reviewers' concerns. Critically, we added the following additional data analyses to the manuscript:

- 1) Single-trial RSA analysis using the best performing decoding model as well as the alternative models as suggested by Reviewers #2 and #3. The analysis based on the best performing model is reported in the main manuscript and other analyses in Supplementary material.
- 2) A decoding analysis on the perirhinal cortex ROI as suggested by Reviewer #2. This is reported in the main text.
- 3) Within-category single-trial decoding analyses as suggested by Reviewer #2, which are now reported in the main text.
- 4) Two additional analyses on non-presented clues as suggested by Reviewer #3. These are reported in the main text and Figure 2.

Reviewer #2 additionally asked us to analyze the results for each presented clue separately (see Comment 2.3). Unfortunately, owing to the time-resolution of fMRI and the fast-paced rate of presentation this analysis is not feasible using the current experimental paradigm. However, we address this concern by discussing this limitation in our study in the main text.

We sincerely believe these analyses and other changes suggested by the three reviewers have significantly improved and strengthened the manuscript. Below, we detail each of the Reviewers' comments, followed by corresponding amendments to the manuscript.

Reviewer #1 (Remarks to the Author):

Comment 1.1. *Error bars are essential in any report of scientific data. Data for individual participants should also have error bars in Fig. 2.*

Response 1.1. The data for individual participants are not summary measures but single data points (accuracy scores) which do not have a meaningful standard deviation. We now report this more carefully in the manuscript.

Previous version Figure 2 legend: Decoding performance using different models in the single-trial analyses. Each line represents a participant. ...

Current version Figure 2 legend: For each model, the raw data points are indicated as a jittered scatter plot on the left side of the summary boxplot. Here, each dot represents the decoding accuracy across all pairwise-classifications ($n = 64\ 620$) for one participant using a given model.

Comment 1.2. *Lots of other numbers are provided in the text without the corresponding measures of variation (e.g. behavior of 93.3%, among many others, won't list them all in here).*

Response 1.2. We apologize for the inconsistency in reporting these values. Measures of variation (standard deviation, minimum and maximum values) have now been added in the text:

Previous (page 6, paragraph2): Over all trials, the participants guessed the implied identities of the target objects at a high accuracy (93.3 percent correct; see also Table S1).

Current (page 6, paragraph 3): Over all trials, the participants guessed the implied identities of the target objects at a high accuracy (mean = 93.3 %, SD = 3.1 %, min = 87.3 %, max = 98.0 %; see also Table S1).

Previous (page 8, paragraph 2): Decoding across semantic category (e.g., elephant vs. car) was expectedly more accurate (mean = 94.2 %, SD = 5.0; $p < 0.05$) than decoding within a semantic category (e.g., elephant vs. giraffe; mean = 64.5%, SD = 7.1; $p < 0.05$).

Current (page 8, paragraph 2): Decoding across semantic category (e.g., elephant vs. car) was expectedly more accurate (mean = 94.2 %, SD = 5.0 %, min = 81.1 %, max = 99.3 %; $p < 0.05$) than decoding within a semantic category (e.g., elephant vs. giraffe; mean = 64.5%, SD = 7.1 %, min = 49.8 %, max = 74.0 %; $p < 0.05$).

Previous (page 18, paragraph 1): Thus, the sample consisted of 17 individuals (mean age = 21.1 years; SD = 3.4 years; mean education = 12.8 years, SD = 1.7 years; ...

Current (page 23, paragraph 1): Thus, the sample consisted of 17 individuals (mean age = 20.9 years; SD = 3.3 years, min = 18 years, max = 31 years; mean education = 12.4 years, SD = 1.5 years, min = 12 years, max = 18 years; ...

Previous (page 19, paragraph 3): The two measurement days were on average 10 days apart (mean = 9.9, SD = 7.9).

Current (page 24, paragraph 4): The two measurement days were on average 10 days apart (mean = 9.9, SD = 7.9, min = 6 days, max = 35 days).

Comment 1.3. *It is not clear exactly how the algorithm is trained and tested. Specifically, do the training data and test data correspond to different presentations of identical cue stimuli? Or can the authors decode information across different cues (e.g. train on completely non-overlapping sets of cues from those used in the test set)?*

Response 1.3. We apologize for the lack of clarity. The imaging data used in training and test never corresponded to different presentations of identical stimuli. For example, here are the stimuli used for the six trials in which the target was “elephant”:

Set*	Triplet	Clue 1	Clue 2	Clue 3
1	A	it has legs	it is thick-skinned	it has a long trunk
	B	it is gray in color	it lives in herds	it has tusks
	C	it is a mammal	it has big ears	it has a good memory
2	D	it is a mammal	it is thick-skinned	it has tusks
	E	it is gray in color	it lives in herds	it has a good memory
	F	it has legs	it has big ears	it has a long trunk

* Sets were presented in different measurement sessions on different days

So, while individual cues were repeated, the clue triplets were trial-unique.

In the analysis scheme where data was averaged across the 6 trials that had the same target object, the model was trained using 58 training items and tested with the two left-out objects. Therefore, the items in the test set did not overlap with the targets in the training set. For example, if the targets for the test set were “elephant” and “hammer”, all trials in which the target was either “elephant” or “hammer” were removed from the training set. In the single-trial analysis scheme, the clue triplets were unique across trials. In this case, the model was trained using 358 clue triplets and tested with 2 left-out clue triplets. In the same way as above, the clue triplet in the test set never occurred in the training set.

In order to clarify the method, we have now made modifications to the manuscript:

Added text on page 29, paragraph 3: The trained model can be used to predict the brain activation patterns for of any novel concept outside the training set for which the corpus-derived semantic coordinates are available.

Added text on page 30, paragraph 2: In this analysis, the six repetitions with unique clue triplets for a given target object were averaged together into a single BOLD activation map using stability selection as described below. The zero-shot decoding model was trained by using 58 of the target items and the training data was used to predict the semantic coordinates of the two left-out target items. The training and evaluation process was iterated 1770 times to cover all leave-two-out combinations.

Previous version on page 24, paragraph 2: In the single-trial analysis, no averaging was performed over the six trials of the same target object, but each trial was considered as an isolated event. The brain activation patterns related to each trial were then used to predict the target object or different sets of features (for details, please see section: Semantic space from text corpus data).

Current version on page 30, paragraph 4: In the single-trial analysis, no averaging was performed over the six trials of the same target object, but each trial using a unique clue triplet was considered a separate event. The brain activation patterns related to each trial were then used to predict the target object or different sets of features (for details, please see section: Semantic space from text corpus data). Thus, the zero-shot decoding model was trained on 358 trials and was used to predict the semantic coordinates of the two left-out trials. As the clue triplets in each trial were unique, the training data did not correspond to different presentations of identical stimuli.

Comment 1.4. *Figure 3 uses a nasty convention in the fMRI literature that is more hurtful than useful. Values below an arbitrary threshold are portrayed as gray. The problem is that these thresholds are completely arbitrary. It is important to show the entire scale of values for every voxel (i.e., no thresholds at all, no gray in the brain rendering, every voxel should have a t value and hence a color).*

Response 1.4. As Reviewer #1 suggests, we now report the entire scale of pseudo-*t* values projected on a cortical surface in the new figure. However, we do believe that it is also important to clearly indicate the areas where these values reach statistical significance (see e.g. Poldrack et al., 2008). The significant effects are indicated in oblique colors and non-significant areas in transparent colors. For completeness, we also report these data on a full 3D volume in Supplementary online material:

Poldrack, R.A., Fletcher, P.C., Henson, R.N., Worsley, K.J., Brett, M., Nichols, T.E., 2008. Guidelines for reporting an fMRI study. *Neuroimage* 40, 409–414.

Revised version of Figure 3.

Previous version p. 26, paragraph 2: The correlation maps of each participant were transformed into MNI space and smoothed at 6 FWHM. The resulting normalized and smoothed images of each participant were subjected to a group-level SnPM analysis using variance smoothing of 6 FWHM and 10000 permutations (SnPM13; <http://go.warwick.ac.uk/tenichols/snpm>). Clusters surviving family-wise-error-corrected $p < 0.05$ that are over 10 voxels in size are reported.

Current version p. 32, paragraph 4: fMRI analyses are prone to increased risk of false positives as statistical tests are performed on a very large number of voxels. To deal with this problem, we indicate the pseudo- t values that survive the voxel-level family-wise error (FWE) corrected p -threshold < 0.05 (height threshold: pseudo- $t = 4.82$). In the Supplementary material (<https://neurovault.org/collections/FNVMTHTC/>), we also report the uncorrected pseudo- t maps of the main RSA analysis (see e.g. Poldrack et al., 2008). We further provide the results of alternative RSA analyses, using the remaining models applied in the single-trial decoding analyses.

Comment 1.5. *There is little information on what the objects are, how similar they are, etc. This is important to better understand the leave-two-out performance and how similar those left out objects are to the rest of the objects.*

Response 1.5. We now present a dissimilarity matrix (Figure S1) in the Supplementary material (please also see Figure below) based on the word2vec semantic coordinates of the target words. The figure also details the individual target item labels. Since the target concepts were collected from only four semantic categories, they are relatively close together in the semantic space as compared to many other semantic decoding studies, where the items have been drawn from multiple semantic categories (e.g. Mitchell et al., 2008 and Sudre et al., 2012: 12 categories; Pereira et al., 180 semantic clusters). As it is harder to decode more similar items, this highlights the effectiveness of our experimental design and decoding model.

Figure S1. The similarity of the semantic coordinates of each target item pair based on word2vec.

We refer to the Figure S1 on page 6, paragraph 3: The stimuli in this study consisted of 60 target objects that fell into four semantic categories (animals, fruits/vegetables, tools, vehicles, see also Supplementary Figure S1).

References:

- Mitchell, T.M., Shinkareva, S.V., Carlson, A., Chang, K.-M., Malave, V.L., Mason, R.A., Just, M.A., 2008. Predicting human brain activity associated with the meanings of nouns. *Science* 320, 1191–1195.
- Pereira, F., Lou, B., Pritchett, B., Ritter, S., Gershman, S.J., Kanwisher, N., Botvinick, M., Fedorenko, E., 2018. Toward a universal decoder of linguistic meaning from brain activation. *Nat. Commun.* 9, 963.
- Sudre, G., Pomerleau, D., Palatucci, M., Wehbe, L., Fyshe, A., Salmelin, R., Mitchell, T., 2012. Tracking neural coding of perceptual and semantic features of concrete nouns. *NeuroImage* 62, 451–463.

Reviewer #2 (Remarks to the Author):

I have two prominent issues relating to (1) whether this reflects evidence from semantic pattern completion, and (2) would like to see extra evidence to bolster the claims made.

Major comments

Comment 2.1. *To my knowledge, semantic pattern completion is a novel idea, and on the surface is attractive and intuitive. However, I'm not sure it's actually a useful construct. After all, when we see a visual object and activate a semantic representation, not all aspects of semantics will be visually present – so is this also a case of semantic pattern completion? If so, every study on the activation of semantics from images can be seen as evidence for semantic pattern completion. This is not to undermine the idea, as it must certainly be the case we do this, but I'm not sure this study shows us how this happens.*

Response 2.1. We thank Reviewer #2 for the insightful comment. We agree that there is plenty of evidence from everyday life demonstrating that semantic pattern completion must take place in some shape or form, otherwise we would not be able to understand the things we perceive. Similarly, we agree that picture naming requires semantic pattern completion given that it is impossible to name an object without accessing its meaning at some level (see e.g. Levelt et al., 1992). However, we argue that the empirical neuroimaging evidence of this process is lacking and that our study provides the first step in demonstrating this process using neuroimaging.

In the present study, we explicitly test whether the semantic representation probed by the clues contains more information than was provided as input. That is, we explicitly control the semantic features we give as input and probe the content of the resulting semantic representations. We are not aware of a previous study that controls this explicitly. The results indicate that the content of the resulting representation is richer than one expected if it was based on the initial clue set alone. We claim this would not be possible if the semantic information would not be enriched during the task in the neural networks and, therefore, it provides indirect neuroimaging evidence on pattern completion. In other words, we demonstrate that we can decode material that is incidental to the task (i.e. features that were not presented).

We added the following text to the Discussion (page 20, paragraph 2): Pattern completion of semantic information is a frequent phenomenon in our everyday life. The brain automatically and effortlessly takes advantage of clues in our environment with prior knowledge about the meaning of objects we encounter. Otherwise, we would not be able to make sense of the world around us. The present study aimed to find empirical neuroimaging evidence for this process. Indeed, the present results demonstrate that we can use clues to elicit a representation that contains more information about the object's meaning than what was provided as input. We do this by showing that 1) the highest decoding accuracy was achieved by combining the rich set of features associated with each object and that 2) we could decode features incidental to the task, i.e. features that were never presented.

Comment 2.2. *Theoretical issues aside, I feel that both the task, and the GLM used, might also undermine the evidence being presented in favour of semantic pattern completion. In terms of the task, participants know this is a guessing game and receive three clues. The first 2 clues will be relatively more shared (less distinctive) and so people know roughly what kind of thing it'll be. The third clue is the most distinctive, and so at this point they will guess what it is, and so the neural representation may well reflect the guess, and therefore the target semantics. Therefore, it can be argued the neural representations do not reflect the clues alone, but the target itself. The data presented backs this up, but doesn't this more reveal that the task elicited semantic representations rather than feature/clue representations, rather than telling us how these pieces were used to elicit semantic representations.*

Response 2.2. We would argue that the clue words have semantic representations of their own which follow the same organizational principles as the representations of the target

words we probe. Indeed, this was our rationale for modeling both clues and target words using word2vec.

We interpret the results as showing that the clue representations are activated in the brain (based on the clue1+2+3 giving a fairly good decoding accuracy). As the Reviewer suggest, we were also able to decode target representations with comparably good accuracy. However, the most successful decoding model was that of a wide selection of target-appropriate features, in accordance with the feature-based accounts of semantics where a target representation would be expected to consist of a rich set of features (including, but not limited to the clues).

We do acknowledge the concern of Reviewer #2 in that while our results demonstrate that the features become combined in time, the present data are limited in revealing how this happens. Please see the Response 2.3 for a more detailed response to this concern.

Comment 2.3. *A related issue is with the GLM used, that is based only on the third clue, and so will reflect the guess (given the above). I feel the process of going through clue 1- 3 would be most revealing in terms of how information is combined.*

Response 2.3. We agree with Reviewer #2 and we did indeed give this a lot of thought when we designed and analyzed the experiment. It would be highly interesting to observe the time course of BOLD activation from the first to the third clue. Unfortunately, however, we cannot disentangle clues 1, 2 and 3 owing to the sluggish haemodynamic response in fMRI, and the current paradigm, where the intervals between clues fixed and relatively short (i.e. $1 \text{ s} < 1 \text{ TR}$). The rationale for the short intervals was twofold:

Firstly, data from extensive behavioral piloting guided us into making the clue presentation relatively rapid so as to **best sync the time at which the participants guessed the correct concept**. Otherwise, it would have been difficult estimate on the moment at which the participants understood the meaning of the target concept. In fact, we have found the time-locking to be one of the most critical challenges in terms of experimental design also when using more time-sensitive methodologies to study pattern completion.

Secondly, at the time we designed the experiment, the approach was completely novel and we did not anticipate achieving such a high signal-to-noise ratio (SNR) even at a single-trial

level. Therefore, we included six repetitions of the same target item to ensure sufficient SNR. In order to include many repetitions, we needed to limit the duration of each individual trial.

Naturally, now that we have demonstrated the usefulness of the guessing game paradigm and the relatively good decoding accuracies even at a single-trial level, the next step is to more carefully examine how the information is combined in time. We are currently in the process of using magnetoencephalography to temporally dissociate individual clues from another and track the semantic pattern completion process in real time.

Comment 2.4. *I feel all these issues make it difficult to make claims relating to semantic pattern completion processes (or the how), rather than an analysis of the result of semantic pattern completion (i.e. a complete semantic representation).*

Response 2.4. We thank Reviewer #2 for pointing out this limitation in our study. It is true that we cannot observe semantic pattern completion in action but we deduce that indirectly. We now explicitly address this issue in the discussion of the manuscript

Added text on page 20, paragraph 2: We suggest that these results reflect the spread of activation in the neural networks as suggested by computational accounts on semantics (Gardner-Medwin, 1976; Hunsaker and Kesner, 2013; Marr, 1971; McClelland et al., 1995). It should be noted, however, that although we deduce that this is most likely the case, we cannot directly observe the pattern completion process with the current design owing to the sluggish haemodynamic response. Future studies are needed to demonstrate the actual process of semantic pattern completion in the brain. We believe that the present methodology (guessing game, word2vec and zero-shot decoding) combined with more time-sensitive brain imaging techniques will be a very fruitful approach in understanding the neural dynamics of that process.

Comment 2.5. I think the data analysis presented is clear and appears robust, however I felt some additional analyses are warranted. In terms of the decoding analysis, it would be useful to know the accuracies for within and between category decoding (or, how specific, semantically, are these predictions). If it is truly a complete semantic representation, rather than at a superordinate category level, then you should see significant within-category effects.

Response 2.5. Regarding the analyses on the averaged data, we provided the decoding accuracies for each target item pair in a confusion matrix in the Supplementary material (Figure S1 in the original manuscript, Figure S2 in the current version). For the present revision, we have included also a confusion matrix on the single-trial data in the Supplementary material (Figure S3). We now also report the within-category decoding results in the main text.

Added text page 10, paragraph 2: For this model, decoding across semantic category was significant in all 17 participants (mean = 77.2 %, SD = 6.3 %, min = 64.4 %, max 87.3 %). Decoding within semantic categories was significant in 16 out of 17 participants (mean = 58.1 %, SD = 3.0 %, min = 52.6 %, max = 63.9 %). Within an individual semantic category, the highest decoding accuracy was achieved for tools (mean = 61.7 %, SD = 6.0 %, min = 49.8 %, max = 72.5 %; significant decoding accuracy in 16/17 participants), followed by vehicles (mean = 60.8 %, SD = 5.1 %, min = 51.4 %, max = 72.4 %; significant decoding accuracy in 15/17), animals (mean = 56.2 %, SD = 3.4 %, min = 50.7 %, max = 65.7 %; significant decoding accuracy in 14/17 participants) and fruits and vegetables (mean = 53.7 %, SD = 4.3 %, min = 46.2 %, max = 61.7 %; significant decoding accuracy in 9/17 participants). These results suggest that the semantic coordinates were sufficiently detailed to distinguish items within the same semantic category.

Figure S3. The single-trial confusion matrix based on leave-two out classifications. The color-scale represents the number of misclassifications in decoding for each trial pair.

We refer to this figure on page 10, paragraph 2: The best performing model was the all available features model, i.e. the one where the resulting semantic coordinates incorporated the combination of all CSLB features of a target object (see also Supplementary Figure S3 for a confusion matrix).

These results demonstrate that the model captures item-level semantic representations and not merely the superordinate level. We would like to also note that it is particularly impressive that the zero-shot decoding works so well as we have only four semantic categories in our data set whereas previous decoding studies have used a far more diverse set of items (e.g. Mitchell et al., 2008 and Sudre et al., 2012: 12 semantic categories; Pereira et al., 180 semantic clusters). Moreover, living things such as animals and fruits/vegetable, which make up 50% of our items, are known to be particularly semantically confusable. Yet, the within-category decoding is somewhat successful also in those categories.

References:

- Mitchell, T.M., Shinkareva, S.V., Carlson, A., Chang, K.-M., Malave, V.L., Mason, R.A., Just, M.A., 2008. Predicting human brain activity associated with the meanings of nouns. *Science* 320, 1191–1195.
- Pereira, F., Lou, B., Pritchett, B., Ritter, S., Gershman, S.J., Kanwisher, N., Botvinick, M., Fedorenko, E., 2018. Toward a universal decoder of linguistic meaning from brain activation. *Nat. Commun.* 9, 963.
- Sudre, G., Pomerleau, D., Palatucci, M., Wehbe, L., Fyshe, A., Salmelin, R., Mitchell, T., 2012. Tracking neural coding of perceptual and semantic features of concrete nouns. *NeuroImage* 62, 451–463.

Comment 2.6. *Further, given the strong hypothesis about the role of the PRC, something like an additional decoding analysis restricted to the PRC would be very compelling.*

Response 2.6. We performed a decoding analysis restricted on the bilateral PRC's as per the Reviewer's request. We now report the results of this analysis in the manuscript.

Added text on page 13, paragraph 1: We conducted an additional region-of-interest (ROI) decoding analysis where we restricted the analysis to the bilateral PRC, given our *a priori* hypothesis regarding the importance of the PRC in combining the features together into object representations. For this region-of-interest analysis, we averaged the BOLD signal in the bilateral PRC across the six repetitions of the same target object and used the semantic coordinates constructed from all available features. This analysis resulted in a statistically significant decoding accuracy in 15 out of 17 participants (mean accuracy = 69.1 %, SD = 6.0 %, min = 59.5 %, max = 79.7 %). This result indicates that the meaning of the target objects can be decoded based on the BOLD signal in the PRC alone. The result further supports the hypothesis that the PRC is involved in decoding object meanings that are reconstructed from a limited set of clues.

Added text on page 18, paragraph 1: A region-of-interest analysis also demonstrated that we could decode the identities of objects based on the BOLD response from PRC alone (significant decoding accuracy in 15/17 participants).

Added text starting from page 33, paragraph 2:

Region of interest analysis

The PRC ROI was based on FreeSurfer's (<https://surfer.nmr.mgh.harvard.edu>) probabilistic PRC label (Augustinack et al., 2013). This label encompasses the medial bank of the collateral sulcus which corresponds to the Brodmann's cytoarchitectonic field 35, i.e. the transentorhinal cortex (Taylor and Probst, 2008; see also Insausti et al., 1998; Kivisaari et al., 2013). The surface-based labels were converted to volume-based ROIs after which the resulting ROI was manually inspected. When necessary, the ROIs were corrected manually such that they continuously covered the entire medial bank of the collateral sulcus.

The BOLD activation maps inside the PRC ROIs were averaged across the six repetitions of each target object. The voxel-wise BOLD signals in the left and right hemispheres were then concatenated resulting in a matrix with 60 rows (number of target items) and n columns, where the n corresponds to the total number of voxels in the left and right PRC ROI. These data were subjected to the zero-shot decoding scheme and used to predict the semantic coordinates of the 60 items. The semantic coordinates were built using the all available features model (i.e. all available features in the CSLB norm data, Devereux et al., 2014).

Comment 2.7. *I also thought the RSA searchlight analyses were very brief. The only analysis presented is based on the target items, but couldn't this be supplemented with either (a) a comparison between RSA of the target words vs. all available features, or (b) a single trial RSA analysis to match the conditions presented in Figure 2.*

Overall, I feel the decoding analysis is aimed at telling us what the nature of the distributed representations are, and the RSA analysis to add spatial specificity. But because the two present different analyses they are a little tricky to combine. I think either expanding the decoding or the RSA would help do this.

Response 2.7. We have made the RSA results more comparable to the decoding analyses as per the Reviewer's suggestion. We now conduct the RSA on a single-trial basis, similarly as

the main decoding analysis. We also provide the RSA analyses for each model used in the main decoding analysis in the Supplementary online material (<https://neurovault.org/collections/FNVMTHTC/>).

Comment 2.8. *A final issue is what does it tell us that the word2vec vectors for the target do a worse job than the combined vectors for all the target's features? Is it that the semantics of the targets are not well estimated by word co-occurrence, but they are by the combined word co-occurrence structure given by their features?*

Response 2.8. We examined the nearest neighbors of the semantic coordinates for the targets and found that they had reasonable representations: the nearest neighbors of all words either belong to the same category as the target word or are meaningful associations or features of the target word (e.g. car -> wheel). The same was true for the all available features model. Therefore, we trust that both semantic coordinate schemes are meaningful.

We postulate that the all available features model performs better than the target model as it combines several target-appropriate features together, and therefore creates a vector representation which explicitly contains information about many relevant features of the concept.

We now address this question also in the Discussion, page 17, paragraph 2: Interestingly, the decoding accuracy was higher for the all available features model, combining word2vec vectors over a rich set of features, than the model using the word2vec vector for the target word only. We speculate that the all available features model performs better than the target model as it combines several target-appropriate features together, and therefore creates a vector representation which explicitly contains information about the relevant features of the concept. This may be a richer representation than that inferred from simple co-occurrence counts of isolated target words or individual clues in the corpus.

Minor issues

Comment 2.9 *Page 11. It wasn't clear to me whether in the RSA analysis the RDM was defined based on word2vec or the target words, or all available features, or just the cosine distance based on feature overlap?*

Response 2.9. We phrased this segment more clearly by making the following amendment:

Previous version, page 11, paragraph 1: We used RSA to determine the brain-wide set of regions in which activation patterns were correlated with the semantic similarity structure among the 60 target objects, as represented by the semantic coordinates of all available features (i.e., the best performing model in the zero-shot decoding). The similarity was defined as the cosine distance between the corpus-derived semantic coordinates

Current version, page 13, paragraph 2: We used a single-trial searchlight RSA to determine the brain-wide set of regions where BOLD activation patterns reflected the semantic similarities of the implied target objects. Here, separately for each searchlight sphere, a representational dissimilarity matrix (RDM) was computed based on the BOLD activation patterns (Kriegeskorte et al., 2008). This RDM reflected the distance ($1 - \text{Pearson's correlation}$) between activation patterns for each pair of trials. We tested for significant correlations between these activation-pattern RDMs and a model RDM which was based on the all available features model, i.e. the best performing model from the zero-shot decoding analysis. Here, the semantic coordinate of each trial was calculated as the sum of semantic coordinates of all available features of the implied target object. The model RDM was computed using the cosine distances between these semantic coordinates.

Amended text on page 32, paragraph 3: The model RDM was based on all available features model. That is, the semantic coordinate of each trial was calculated as the sum of semantic coordinates of all available features of the implied target object (see model 4 above). The resulting model RDM was a 360×360 matrix, where the value in each cell reflects the cosine distance between the semantic coordinates of a pair of trials. The model RDM was compared to activation pattern RDMs which were constructed for each spherical searchlight (radius = 7 mm) across each voxel in the gray matter volume. The activation pattern RDMs were symmetrical 360×360 BOLD matrices, where the value in each cell reflects the dissimilarity ($1 - \text{Pearson's correlation}$) of activation patterns between a pair of trials. A whole-brain correlation map was produced by calculating Spearman's rank correlations between the activity-pattern RDMs and semantic model RDMs. The correlation maps were Fisher

transformed in order to make them normally distributed and projected back onto each searchlight's center voxel.

Comment 2.10. *Figure 3 legend. It would be useful to add thresholds, statistical corrections etc for clarity.*

Response 2.10. We now report the thresholds and statistical correction in the legend: "Pseudo- t values for each voxel are projected onto the surface of a template brain. The vertices where pseudo- t value was not statistically significant (based on a permutation test at a FWE-corrected level of $p < .05$, pseudo- $t > 4.82$) are semi-transparent. "

Comment 2.11. *I wondered if the authors had considered analyses comparing activation during clue 1/2/3 vs the guess, to see how clue representations incrementally reflect the guess (this may get closer to the how question).*

Response 2.11. We did consider this but unfortunately, it is not possible with the current imaging method and design (please see Response 2.3 for details).

Comment 2.12. *Was there a reason the study was conducted over two session?*

Response 2.11. The study was conducted over two sessions in order to maximize the number of repetitions of the target items. Each session lasted approximately 45 minutes + 10 minute MPAGE acquisition. We thought that a single session would have been too strenuous for the participants and jeopardized the quality of the data.

We made the following specification to the text:

Previous (page 19, paragraph 3): The fMRI experiment was conducted on two days, with three measurement sessions (i.e., blocks) on each day. The two measurement days were on average 10 days apart (mean = 9.9, SD = 7.9).

Current (page 24, paragraph 4): The fMRI experiment was conducted on two days, with three measurement sessions (i.e., blocks) on each day. We divided the data acquisition into two separate days to ensure that the participants would be able to sustain attention throughout

the experiment. The two measurement days were on average 10 days apart (mean = 9.9, SD = 7.9, min = 6 days, max = 35 days) with each fMRI measurement lasting ca. 45 minutes in total.

Reviewer #3 (Remarks to the Author):

Comment 3.1. The authors mention pattern completion in relation to episodic memory, citing two largely theoretical papers. There is a very rich literature on hippocampal pattern completion (and neocortical reinstatement), which starts with Marr. Connections with this literature would broaden the scope of the current manuscript, as well as provide a richer empirical and computational background to the study:

Theoretical work:

Marr (1971) Simple memory: a theory of archicortex

Gardner-Medwin (1976) The recall of events through the learning of associations between their parts

McClelland et al (1995) Why there are complementary learning systems in the hippocampus and neocortex

Behavioural work:

Horner & Burges (2014) Pattern completion in multielement event engrams

Vieweg et al. (BioArXiv) Memory Image Completion (MIC): Establishing a task to behaviourally assess pattern completion in humans

fMRI work:

Wheeler et al. (2000) Memory's echo: vivid remembering reactivates sensory-specific cortex

Woodruff et al. (2005) Content-specificity of the neural correlates of recollection

Staresina et al. (2012) Episodic reinstatement in the medial temporal lobe

Horner et al. (2015) Evidence for holistic episodic recollection via hippocampal pattern completion

Review papers:

Hunsaker & Kesner (2013) The operation of pattern separation and pattern completion processes associated with different attributes or domains of memory

Horner & Doeller (2017) Plasticity of hippocampal memories in humans

Response 3.1. We thank Reviewer #3 for these suggestions. We now provide a more comprehensive background to pattern completion literature. We added the following paragraphs to the text:

Page 4, paragraph 1: In this study, we define pattern completion as a partial clue leading into the retrieval of a previously learned memory trace, and the neural basis of this process has been under intensive scrutiny for decades (e.g. Marr, 1971; O'Reilly and McClelland, 1994; O'Reilly and Rudy, 2001; Treves and Rolls, 1994). The research has primarily centered around the recall of episodic memories and the role of the hippocampus in this process. Several studies have provided also neuroimaging (fMRI) evidence on the role of the hippocampus in binding partial cues with the context in which they were learned (Bakker et al., 2008; Horner et al., 2015; Staresina et al., 2012; Wheeler et al., 2000; Woodruff et al., 2005). Pattern completion has been demonstrated for example in the visual domain (Tang et al., 2014), and it has been suggested that pattern completion takes place also for other types of information, including semantic memories (Hunsaker and Kesner, 2013). However, there is little we know about the neural underpinnings of pattern completion of semantic memories.

Page 19, paragraph 3: Based on the present findings, we postulate that the neural basis of semantic pattern completion differs from that of episodic memories, which has been attributed to the hippocampus (Bakker et al., 2008; Gardner-Medwin, 1976; Horner et al., 2015; Horner and Burgess, 2014; Hunsaker and Kesner, 2013; Marr, 1971; McClelland et al., 1995; Staresina et al., 2012; Woodruff et al., 2005). Specifically, we suggest that the pattern completion of semantic information about objects takes place in the ventral stream and the PRC. We suspect that this process is largely independent of the hippocampal system, as semantic knowledge about objects has been acquired relatively early in life and over a long period of time (McClelland et al., 1995). In other words, the recall of semantic information does not require taking into account newly acquired information that would place demands on a more adaptive system in the hippocampus. Instead, semantic pattern completion may require disambiguation of feature combinations (i.e. the clues in the present task) and, therefore, rely on the complex representations of feature conjunctions provided by the PRC (Bartko et al., 2007; Bussey et al., 2005). Thus, pattern completion of semantic object memories may take place in the same system that is involved in processing and perceiving objects.

Comment 3.2. *I found the “zero-shot learning and model evaluation” in the methods difficult to follow. Perhaps clearer statements regarding what the training data are and what the test data are, as well as what this analysis is designed to provide evidence for, would help the reader.*

Response 3.2. Please refer to the changes we made in response to Comment 1.3.

We added the following text on page 29, paragraph 2: The aim of these analyses was to test whether we can establish a statistically significant mapping between the brain activation patterns and the word2vec semantic coordinates.

Page 31, paragraph 3: In the last step, we wanted to test which one of the trained models (i.e. using different sets of semantic coordinates as described above) provided the best mapping to the observed brain activation patterns. To this aim, the decoding accuracies between different models were compared using a pairwise *t*-test using a Bonferroni correction.

Comment 3.3. The analysis comparing the different models provides evidence that the “all available features” model gives the highest classification accuracy. To what extent might this result be driven by the increased number of features, rather than true pattern completion? For example, if presented with cues for an elephant, on one trial I might bring to mind that they are endangered (non-cued) and on another that they have big ears (non-cued). The “all available features” model would work best here, but it doesn’t mean I’m always pattern completing. Instead, I’m bringing to mind different non-cued features on each trial. I wonder whether a model selecting 3 (random) non-cued features could help here. At the least, it would control for the overall number of features (relative to the “clue 1+2+3” model), and would also show you can classify solely on the basis on non-cued features.

Response 3.3. We thank Reviewer #3 for this important comment. We have now added two additional zero-shot decoding models which help to clarify this issue. In one model, we mix the clues used to evoke concepts across blocks. We mixed them in such a way that the clues used to predict the brain activation patterns are never the same as those presented to the participant (e.g. for a trial where we would probe “elephant” using clues “has legs”, “is thick-skinned” and “has a long trunk”, we would decode the brain activation patterns using features “gray”, “herd”, “tusk”, i.e. clues from another block). This way we can test whether

we can decode the target objects using clues that were never presented in a given trial. This analysis yielded a significant decoding accuracy in all but one participant. In the second analysis, we included all clues from the CSLB norm data that were never presented in the guessing game task (akin to the all available features but excluding the nine clue features used in the guessing game task). This analysis yielded a significant decoding accuracy in all 17 participants. We added the following paragraphs to the text:

Page 9, paragraph 2: In addition, we generated two models which excluded those clues that were explicitly provided to probe the target concept. In one variant (5), we mixed the clue sets across blocks such that the semantic coordinates of a given trial were constructed using the clue features of another trial with the same target object (“mixed clues”). This way, the clues used to predict the brain activation patterns are not the same as those actually presented to the participant (e.g. for a trial where we would probe elephant using clues “has legs”, “is thick-skinned” and “has a long trunk”, we would decode the brain activation patterns using features “gray”, “herd”, “tusk”, i.e. clues from another block). In the last model (6), we included all features from the CSLB norm data that were not presented as clues (“all nonclues”; akin to the all available features model above, but excluding the nine clue features used in the guessing game task). For all these models, we established a semantic coordinate using the word2vec model and applied the same procedure as for the target words (Figure 2).

Page 16, paragraph 3: Further evidence for semantic pattern completion was offered by models incorporating features that had not been presented to the participant. We were able to decode brain activation patterns using non-overlapping clues from other trials (above chance-level decoding for 16 out of 17 participants). Moreover, a high level of decoding accuracy could also be achieved by using a rich set of target features (akin to the all available features model) while excluding those nine features that were used as clues in the experiment. Indeed, this analysis performed significantly above chance level for all participants and better than the other models, surpassed only by the model that incorporated all available features including the nine features used as clues in the experiment. This means that the guessing game task activated features that were incidental to the task, i.e. those that were not given as input in the experiment. We suggest this finding provides further support for the notion of pattern completion in the current study.

Page 10, paragraph 3: In the next step, we attempted to directly test whether the task elicited features that were not explicitly presented (i.e. whether pattern completion took place). We focused on the two models that included other features of the target objects than the ones that were explicitly provided in a specific trial (Figure 2; i.e., mixed clues (model 5) and all nonclue features (model 6)). In the mixed clues model, there was no overlap between the clue features presented to the participant and those used to create the word2vec semantic coordinates. Using this model, the overall decoding accuracy was significant for all but one participant (see Figure 2). When the model combined all available features from the CSLB dataset *excluding* the nine clues used in the task, a significant decoding accuracy was obtained for all 17 participants (see Figure 2). These results indicate that we can reach a significant decoding accuracy even when the explicitly presented clues are excluded from the model.

Page 27, paragraph 2: In addition, we generated two models which excluded the clues used to probe the target concept: (5) in one of these models, we mixed the clue sets across blocks such that the semantic coordinates of a given trial were constructed using the clue features of another trial with the same target item. This way, the clues used to predict the brain activation patterns were never the same as those that had been presented to the participant (e.g. for a trial where we would probe elephant using clues “has legs”, “is thick-skinned” and “has a long trunk”, we would decode the brain activation patterns using features “gray”, “herd”, “tusk”, i.e. clues from another block). In the final model (6), we calculated the sum of semantic coordinates of all such features that were not presented in the guessing game task for a given target concept (akin to the all available features model but excluding the nine clue features used in the guessing game task).

Amended Figure 2. which now incorporates all six models.

Comment 3.4. *The RSA analysis only uses the “all available features” model. Did the authors also perform the analyses for the other models? Do they produce largely similar (but somewhat weaker) results, or are there any potentially interesting dissociations? For example, I might imagine that the “target” model might work well further up the ventral visual stream (e.g., PRC) but not in more posterior ventral visual regions.*

Response 3.4. We thank Reviewer 3 for this suggestion. We now provide single-trial RSAs on all models used in the decoding analysis, in the effort to make these two sets of analyses more comparable (please also see our Response 2.7 to Reviewer 2). All four models produce very similar results in terms of ventral stream involvement. The models differ primarily on the extent of the significant clusters, which does not give grounds to suspect there are

anatomical dissociations between these models. We now provide these results in the online Supplementary methods (<https://neurovault.org/collections/FNVMTHTC/>).

We added the following text on page 19, paragraph 2: We note that the RSAs yielded largely overlapping anatomical patterns for all models tested in this study (i.e. clue 3, mixed clues, clue 1 + 2 + 3, target word, all nonclues, all available features; see Supplementary online material: <https://neurovault.org/collections/FNVMTHTC/>). Thus, despite the fact that we found significant differences in decoding accuracy in the zero-shot decoding analysis, all six models had similar anatomical patterns of correlation with BOLD activity which differed primarily in extent. Indeed, we would like to emphasize the fact that each one of the six models contained relatively high-level conceptual information about objects (i.e. as compared to e.g. low-level perceptual features). We propose that this explains the overlapping patterns of activity in regions associated with semantic object processing.

Reviewers' comments:

Reviewer #1 (Remarks to the Author):

Reconstructing meaning from bits of information

The authors have made multiple improvements in the presentation.

The new version of Figure 3 reveals that there are lots of brain regions that seem to be activated according to the searchlight results. Arbitrary significance thresholds might make the results appear to be localized but Nature (the real one, not the journal), does not care about thresholds.

The discussion of what is in the training set and in the test set still remains unclear. The key question is whether there is anything about semantics in here. In the single trial analyses (arguably the most critical ones), it seems based on the responses that there is a lot of commonality between the training set and the test set. Even if the exact same triplet is not shown in both, a lot of the words are, it seems, hence it is not clear that there is any extrapolation here. In the case where data are averaged across the 6 stimuli, it is also not clear how much was in common between the training set and test set. The authors should be commended for tackling a fascinating and very difficult question. To make serious claims about semantics, it is critical to ensure that semantics is the only variable over which the decoder needs to extrapolate and that there are no other confounds of similarity across stimuli. Even though the results are not convincing, this is certainly a step in the right direction to study a very important and challenging question in the field.

Reviewer #2 (Remarks to the Author):

Kivisaari and colleagues have addressed many of my concerns and I appreciate the new analyses presented, which improves the manuscript. The main concern that remains is the nature of some claims the study makes.

I'm aware I'm being a little bit picky, but in various places (including the abstract and final conclusions) the paper claims to have shown how information is combined, and how pattern completion occurs. To me, this is a mechanistic claim, whereby the study will have shown us some process by which pattern completion takes place. However, I don't think the data presented does this. This is something I raised in the initial review, and note the authors have addressed this somewhat. However, there are still claims made about how information is combined in the paper. Rather than explaining how, the study shows that when people guess a concept, they activate the full semantic representation of that concept, which is represented across a distributed network in frontal, temporal and parietal regions. This is consistent with the vast majority of the paper, but I would suggest the authors withhold from saying the study revealed a mechanism, when we only observe the result of it.

Other minor points:

Page 18, Typo: Other regions include the bilateral retrosplenial cortex, which used a single-trial searchlight RSA to determine which in previous research has been associated with visual imagery and memory

Page 19: Here you argue that semantic pattern completion is independent of the hippocampus. My comment here is not a theoretical one, as I'm not challenging this, but based on your own RSA searchlight maps. Don't they contain hippocampus effects too? From looking at the neurovalut data, it actually looks like one of the peak effects is in the left hippocampus. How does this change your interpretation?

Page 19/20: 'Thus, pattern completion of semantic object memories may take place in the same system that is involved in processing and perceiving objects.' This is the same as claimed for mental imagery of objects, which may link to the fact people are guessing a concept here.

Reviewer #3 (Remarks to the Author):

The authors have responded thoughtfully and thoroughly to my comments. I have no further comments on the manuscript.

RESPONSE TO REVIEWERS

Response to Reviewer #1

Comment 1.1 The discussion of what is in the training set and in the test set still remains unclear. The key question is whether there is anything about semantics in here. In the single trial analyses (arguably the most critical ones), it seems based on the responses that there is a lot of commonality between the training set and the test set. Even if the exact same triplet is not shown in both, a lot of the words are, it seems, hence it is not clear that there is any extrapolation here. In the case where data are averaged across the 6 stimuli, it is also not clear how much was in common between the training set and test set. The authors should be commended for tackling a fascinating and very difficult question. To make serious claims about semantics, it is critical to ensure that semantics is the only variable over which the decoder needs to extrapolate and that there are no other confounds of similarity across stimuli. Even though the results are not convincing, this is certainly a step in the right direction to study a very important and challenging question in the field.

Response 1.1. We apologize for not explaining clearly the relationship between training data and test data in our previous Response to Reviewers letter. We also thank Reviewer #1 for bringing up this important issue since it made us realize a critical typo we had in the Methods section which served to add to this confusion. Below, we explain the details of the zero-shot design, with the focus on the intermediate features used in the decoding and on the relationship between the training and test sets. We believe this will resolve the concern of Reviewer #1.

In our previous response, we detailed the nature of the word clues used to probe the target concepts in the experiment. We realize now that our explanation was misleading since the clue words presented in the experiment were never used to train the zero-shot decoding model as such. Instead, we used word2vec and corpus-derived semantics to extract a set of intermediate features (i.e. 300-dimensional semantic coordinate) for each of the items, whether these be targets of the single trials or targets averaged across six trials. Depending on the model being tested, the word2vec was applied to the target word, the clue words or other feature words extracted from an independent norm data set. When the model included more than one word, we summed the resulting word2vec coordinates. **In the end, each item in the zero-shot decoding analysis was represented with a single point (i.e. semantic coordinate) in a 300-dimensional semantic space.**

For training, the semantic coordinates and fMRI BOLD responses were paired, per item, to establish a mapping between the semantic space and brain activation patterns. The training was performed such that the model was exposed to a subset of the items (i.e. the paired semantic coordinate and BOLD activation pattern of each trial/average of trials). **The training set always excluded the two test items. In the single-trial analysis, we additionally excluded from training the 5 + 5 trials representing the same target concept as the test items** (i.e. the trials the Reviewer noted shared a lot of commonalities). This way, we also ensured the semantic coordinates of the targets being predicted never appeared in the training set. Please also see the amendment to the main text below.

In the testing stage, the mapping was evaluated using the two test items which were excluded from the training set. This process was iterated such that all of the leave-two-out combinations, across all items, were covered. The mapping from the training stage was based entirely on items that did not contain the same implied target concept as the test items. **This way, the mapping was always applied to two novel semantic coordinates and two novel patterns of brain activation that had never appeared in the training set.** This approach therefore follows the same principles as previous studies using the zero-shot decoding approach (e.g. Mitchell et al., 2008, Sudre et al., 2012).

We apologize for a critical typo in the text which may also have led the Reviewer to suspect the training and test data were overlapping. Namely, the training data in the single-trial analysis consisted of 348 items and not 358 items. We have now corrected this typo and clarified the relationship of the training and test data in the main text:

Previously on page 31, paragraph 1: In the single-trial analysis, no averaging was performed over the six trials of the same target object, but each trial using a unique clue triplet was considered a separate event. The brain activation patterns related to each trial were then used to predict the target object or different sets of features (for details, please see section: Semantic space from text corpus data). Thus, the zero-shot decoding model was trained on 358 trials and was used to predict the semantic coordinates of the two left-out trials.

In the classification stage, we ignored those pairwise leave-two-out combinations that would have included the same target object from different trials (e.g., dog block 1 vs. dog block 2, hammer block 2 vs. hammer block 4). Note that we did not use stability selection in the single-trial analysis, since there were no repeated trials over which stability selection could sensibly

have been performed. Furthermore, as each trial had a different set of clues, we did not want to potentially wipe out this variability.

Currently on page 31, paragraph 1:

In the single-trial analysis, no averaging was performed over the six trials of the same target object, but each trial using a unique clue triplet was considered a separate item. The brain activation patterns related to each trial were then used to predict the semantic coordinates (for details, please see section: Semantic space from text corpus data). First, the test pair was selected after which the other 5 + 5 trials corresponding to the same target concepts were removed from the training set. Thus, the zero-shot decoding model was trained on 348 trials, i.e. all twelve trials representing the two targets we tried to predict were excluded from the training set. Note that we did not use stability selection in the single-trial analysis, since there were no repeated trials over which stability selection could sensibly have been performed. Furthermore, as each trial had a different set of clues, we did not want to potentially wipe out this variability.

Finally, as regards the Reviewer's concern that the words might be processed only on a superficial level (i.e. that there is no semantic analysis involved), we believe the best evidence is provided by the main analyses: here we construct the semantic coordinates (i.e. intermediate features in the zero-shot decoding analysis) such that we include information that was never explicitly presented to the participant ("All available features" model). This model results in significantly better training/prediction performance than using semantic coordinates solely based on the clue words ("Clue 1 + 2 + 3" model). In particular, we point to the two additional analyses we provided in the last revision, i.e. the "Mixed clues" and the "All nonclues" models, which demonstrate that we can also successfully decode the neural representations based on features that were not presented as clues.

References:

Mitchell, T.M., Shinkareva, S.V., Carlson, A., Chang, K.-M., Malave, V.L., Mason, R.A., Just, M.A., 2008. Predicting human brain activity associated with the meanings of nouns. *Science* 320, 1191–1195. <https://doi.org/10.1126/science.1152876>

Sudre, G., Pomerleau, D., Palatucci, M., Wehbe, L., Fyshe, A., Salmelin, R., Mitchell, T., 2012. Tracking neural coding of perceptual and semantic features of concrete nouns. *NeuroImage* 62, 451–463. <https://doi.org/10.1016/j.neuroimage.2012.04.048>

Response to Reviewer #2

Comment 2.1. ...in various places (including the abstract and final conclusions) the paper claims to have shown how information is combined, and how pattern completion occurs. To me, this is a mechanistic claim, whereby the study will have shown us some process by which pattern completion takes place. However, I don't think the data presented does this. This is something I raised in the initial review, and note the authors have addressed this somewhat. However, there are still claims made about how information is combined in the paper. Rather than explaining how, the study shows that when people guess a concept, they activate the full semantic representation of that concept, which is represented across a distributed network in frontal, temporal and parietal regions. This is consistent with the vast majority of the paper, but I would suggest the authors withhold from saying the study revealed a mechanism, when we only observe the result of it.

Response 2.1. We have now changed the wording of the text to reflect the fact that we cannot observe the process of pattern completion (i.e. a mechanistic account), but the end result.

Page 2 (Abstract) modified text. We can easily identify a dog merely by the sound of barking or an orange by its citrus scent. In this work, we ~~study the neural underpinnings of how the~~ aim to demonstrate using neuroimaging that the brain combines bits of information into meaningful object representations.

...

We conclude that our experimental protocol allowed us to ~~observe how~~ demonstrate that the brain completes fragmented information into rich object meanings and identify brain regions supporting the resulting representations of objects and ~~suggest neuroanatomical underpinnings for this process.~~

Previously on page 3, paragraph 1: Thus, in this study, we ask whether semantic pattern completion can be demonstrated in the human brain and what brain regions are involved in integrating features together into rich representations of objects.

Currently on page 3, paragraph 1: Thus, in this study, we ask whether we can demonstrate that semantic pattern completion occurs in the human brain and identify the brain regions which support the resulting rich representations of objects.

Previously on page 6, paragraph 2: We further employ representational similarity analysis (RSA, Kriegeskorte et al., 2008) to visualize the brain regions which are involved in representing the target objects or in completing the patterns of object features into target concepts.

Currently on page 6, paragraph 2: We further employ representational similarity analysis (RSA, Kriegeskorte et al., 2008) to visualize the brain regions which are involved in representing the implied target objects in the guessing game task.

Previous version on page 21, paragraph 2: The present neuroimaging study used a novel experimental design to examine how the brain completes patterns of fragmented information into meaningful, coherent semantic representations. This design, coupled with our machine learning models, allowed us to study, for the first time, how the brain takes advantage of very limited information and enriches it with prior knowledge of object meaning. The present results give strong support for the distributed, feature-based models of semantics in the brain and suggest that the ventral stream is involved in binding the features together into coherent object representations.

Current version on page 21, paragraph 2: The present neuroimaging study used a novel experimental design to demonstrate that the brain completes patterns of fragmented information into meaningful, coherent semantic representations. This design, coupled with our machine learning models, allowed us to show, for the first time using neuroimaging, that the brain takes advantage of very limited information and enriches it with prior knowledge of object meaning. The present results give strong support for the distributed, feature-based models of semantics in the brain and suggest that rich representations of object meanings are partly supported by the ventral stream and the PRC.

Comment 2.2. Page 18, Typo: Other regions include the bilateral retrosplenial cortex, wh used a single-trial searchlight RSA to determine ich in previous research has been associated with visual imagery and memory

Response 2.2. We corrected the typo.

Comment 2.3. Page 19: Here you argue that semantic pattern completion is independent of the hippocampus. My comment here is not a theoretical one, as I'm not challenging this, but based on your own RSA searchlight maps. Don't they contain hippocampus effects too? From looking at the nerovlut data, it actually looks like one of the peak effect is in the left hippocampus. How does this change your interpretation?

Response 2.3. We acknowledge this point and agree that our wording is troublesome. The extent to which the cluster falls on the hippocampus depends on the template brain used in visualization, which reflects the highly variable individual anatomy of the medial temporal lobe region. To avoid any misconceptions, we have changed the wording of this paragraph.

Previous version page 19, paragraph 3: Specifically, we suggest that the pattern completion of semantic information about objects takes place in the ventral stream and the PRC. We suspect that this process is largely independent of the hippocampal system, as semantic knowledge about objects has been acquired relatively early in life and over a long period of time (McClelland et al., 1995). In other words, the recall of semantic information does not require taking into account newly acquired information that would place demands on a more adaptive system in the hippocampus. Instead, semantic pattern completion may require disambiguation of feature combinations (i.e. the clues in the present task) and, therefore, rely on the complex representations of feature conjunctions provided by the PRC (Bartko et al., 2007; Bussey et al., 2005).

Current version version page 19, paragraph 3: Specifically, we suggest that the pattern completion of semantic information about objects partly takes place in the ventral stream and the PRC. We suggest that semantic pattern completion may require disambiguation of feature combinations (i.e. the clues in the present task) and, therefore, rely on the complex

representations of feature conjunctions provided by the PRC (Bartko et al., 2007; Bussey et al., 2005).

Comment 2.4. Page 19/20: 'Thus, pattern completion of semantic object memories may take place in the same system that is involved in processing and perceiving objects.' This is the same as claimed for mental imagery of objects, which may link to the fact people are guessing a concept here.

Response 2.4. We agree. We made the following amendment:

Page 20, paragraph 1. It should be noted that similar effects have been demonstrated for mental imagery (see e.g. O'Craven and Kanwisher, 2000) and that in the current framework we cannot dissociate the process of pattern completion from the end result of mental imagery (i.e. the target concept). Therefore, further research is required to provide a mechanistic account of the process of semantic pattern completion in the brain.

REVIEWERS' COMMENTS:

Reviewer #1 (Remarks to the Author):

The authors have made improvements to the manuscript.

I am not quite convinced that there is any semantic information in here, due to the overlap between the training and test sets. However, again I would like to commend the authors for tackling a fascinating and very difficult question.

Also, the authors did not address the first question in the review of the revised version, copied and pasted again here:

"The new version of Figure 3 reveals that there are lots of brain regions that seem to be activated according to the searchlight results. Arbitrary significance thresholds might make the results appear to be localized but Nature (the real one, not the journal), does not care about thresholds.

"

Despite these issues, I think that there are interesting questions and analyses in this study and that it is worth publishing the work so that the scientific community can discuss it and evaluate it.

It's too bad that the author make so many excuses to share the data. Sharing all code and data publicly is the best way of validating all results for the scientific community.

RESPONSE TO REVIEWERS REQUESTS

Request 1. Regarding the comment of Reviewer #1 (see end of email) "The new version of Figure 3 reveals that there are lots of brain regions that seem to be activated according to the searchlight results. Arbitrary significance thresholds might make the results appear to be localized but Nature (the real one, not the journal), does not care about thresholds", I suggest that this should be addressed by adding a paragraph in the discussion about the possible role of areas in which significance was not quite reached.

Response 1. We added a paragraph in the discussion where we discuss the possible role of subthreshold regions:

"The selection of a statistical threshold is to some extent arbitrary. As illustrated in Figure 3, there were multiple regions where correlations failed to reach the selected statistical threshold, but which may nonetheless be functionally linked with itemlevel semantic processing of objects. The full range of effects can be seen in the uncorrected statistical images which we have made available <https://aaltoimaginglanguage.github.io/guess/>). These data demonstrate that effects which may seem discrete with the chosen threshold in fact reflect a continuum of effects over a larger region."